# HOVERCAT: A novel aerial system for evaluation of aerosol-cloud interactions

Jessie M. Creamean[1,2], Katherine M. Primm[1], Margaret A. Tolbert[1], Emrys G. Hall[1,3], Jim Wendell[3], Allen Jordan[1,3], Patrick J. Sheridan[3], Jedediah Smith[4], Russell C. Schnell[3]

[1]Cooperative Institute for Research in Environmental Sciences, University of Colorado, Boulder, CO, 80309, USA
[2]Physical Sciences Division, National Oceanic and Atmospheric Administration, Boulder, CO, 80305, USA
[3]Global Monitoring Division, National Oceanic and Atmospheric Administration, Boulder, CO, 80305, USA
[4]Smith & Williamson, Corvallis, OR, 97330, USA

*Correspondence to*: Jessie M. Creamean (jessie.creamean@noaa.gov)

**Abstract.** Aerosols have a profound impact on cloud microphysics through their ability to serve as ice nucleating particles (INPs). As a result, cloud radiative properties and precipitation processes can be modulated by such aerosol-cloud interactions. However, one of the largest uncertainties associated with atmospheric processes is the indirect effect of aerosols on clouds. The need for more advanced observations of INPs in the atmospheric vertical profile is apparent, yet most ice nucleation measurements are conducted at the ground or during infrequent and intensive airborne field campaigns. Here, we describe a novel measurement platform that is less expensive and smaller (< 5 kg) when compared to traditional aircraft and tethered balloon platforms and that can be used for evaluating two modes of ice nucleation (i.e., immersion and deposition). HOVERCAT (Honing On VERtical Cloud and Aerosol properTies) flew during a pilot study in Colorado, USA up to 2.6 km above mean sea level (1.1 km above ground level) and consists of an aerosol module that includes an optical particle counter for size distributions (0.38 – 17 µm in diameter) and a new sampler that collects up to ten filter samples for offline ice nucleation and aerosol analyses on a launched balloon platform. During the May 2017 test flight, total particle concentrations were highest closest to the ground (up to 50 cm$^{-3}$ at < 50 m above ground level) and up to 2 in $10^2$ particles were ice nucleation active in the immersion mode (at −23 °C). The warmest temperature immersion and deposition mode INPs (observed up to −6 °C and −40.4 °C, respectively) were observed closest to the ground, but overall INP concentrations did not exhibit an inverse correlation with increasing altitude. HOVERCAT is a prototype that can be further modified for other airborne platforms, including tethered balloon and unmanned aircraft systems. The versatility of HOVERCAT affords future opportunities to profile the atmospheric column for more comprehensive evaluations of aerosol-cloud interactions. Based on our test flight experiences, we provide a set of recommendations for future deployments of similar measurement systems and platforms.

## 1 Introduction

Some of the least understood atmospheric processes are aerosol-cloud interactions, and specifically, those with aerosols that serve as ice nucleating particles (INPs) (Boucher et al., 2013). Formation and microphysical modulation of cloud droplets and

ice crystals is highly dependent upon the types and number of aerosols that serve as cloud condensation nuclei (CCN) and INPs. In the absence of CCN and INPs, clouds would in theory require > 400% humidity and < −36 °C to form droplets and ice crystals, respectively; conditions atypical of mixed-phase clouds (Pruppacher and Klett, 1997). Aerosol-induced microphysical modifications influence cloud lifetime and albedo (Morrison et al., 2005), as well as the production of more or less precipitation, particularly in mixed-phase cloud systems. INPs nucleate ice through pathways dependent upon temperature, saturation with respect to ice, and the INP type (Hoose and Möhler, 2012). The modes of heterogeneous ice nucleation include: 1) condensation freezing whereby ice is formed concurrently with the initial formation of liquid on CCN at supercooled temperatures, 2) immersion freezing whereby an INP is immersed in an aqueous solution or water droplet via activation of CCN during liquid cloud formation, 3) contact freezing whereby an INP approaches the air–water interface of a droplet (e.g., via a collision) and initiates freezing, 4) deposition nucleation whereby ice is formed from supersaturated vapour with respect to ice ($RH_i > 100\%$) on an INP directly, and 5) pore condensation and freezing whereby water vapour is condensed into voids and cavities followed by glaciation (Coluzza et al., 2017; Cziczo et al., 2017; Hoose and Möhler, 2012; Kanji et al., 2017; Marcolli, 2014; Vali et al., 2015).

Immersion freezing is the most relevant to primary ice formation in mixed-phase clouds and requires that INPs initially serve as, or in conjunction with, CCN, whereas deposition nucleation is prevalent in mixed-phase and dominant in cirrus cloud ice formation (Kanji et al., 2017). Aerosols such as mineral dust, soil dust, sea salt, volcanic ash, black carbon from wildfires, and primary biological aerosol particles (PBAPs) have been shown to serve as INPs (Conen et al., 2011; Cziczo et al., 2017; DeMott et al., 1999; Hoose and Möhler, 2012; McCluskey et al., 2014; Murray et al., 2012; Petters et al., 2009). Among these, dust and PBAPs are the most efficient INPs found in the atmosphere (Cziczo et al., 2017; Murray et al., 2012). Dust is the most atmospherically-abundant INP, forming ice as warm as −10 °C, but primarily at temperatures < −15 °C (Hoose and Möhler, 2012; Murray et al., 2012). On the other hand, PBAPs are relatively rare in the atmosphere, but can form ice as warm as −1 °C (Despres et al., 2012; Schnell and Vali, 1976; Vali et al., 1976; Vali and Schnell, 1975). However, constraining aerosol-cloud impacts in models ranging from the cloud-resolving to climate scales, specifically when parameterizing INPs, remains a significant challenge due to limited observations (Coluzza et al., 2017; Cziczo et al., 2017; DeMott et al., 2010).

A number of previous ground-based field measurements dating back to the 1950s have provided noteworthy advancements in understanding the sources and efficiencies of INPs (e.g., Bigg, 2011; Durant et al., 2008; Garcia et al., 2012; Huffman et al., 2013; Jayaweera and Flanagan, 1982; Mason et al., 2015; McCluskey et al., 2014; Mossop, 1963; Murray et al., 2012; Petters et al., 2009; Prenni et al., 2009b; Prenni et al., 2013). Further, previous work has evaluated INP concentrations and at times composition in detritus, soil, water from lakes and oceans, surface microlayers, and precipitation samples to assess INP sources (e.g., Conen et al., 2016; Creamean et al., 2014; DeMott et al., 2016; Hill et al., 2016; Irish et al., 2017; Moffett, 2016; O'Sullivan et al., 2014; Petters and Wright, 2015; Pietsch et al., 2017; Pouzet et al., 2017; Schnell, 1977; Schnell and Vali, 1972, 1973, 1975; Stopelli et al., 2015; Tobo et al., 2014). Analysis of INPs in precipitation samples take a step in the direction

of vertical profiling of INPs, making the assumption that the INPs in precipitation are what initiated ice formation in the clouds above; however, there are caveats associated with artefacts from scavenging during raindrop or snowflake descent, aerosolization methods, and redistribution of residue particles in collected liquid precipitation samples (Creamean et al., 2014; Hanlon et al., 2017; Petters and Wright, 2015).

Although observations at the ground afford detailed information regarding the characterization of INP sources, they may not be representative of INPs in the atmospheric column, where they have the direct ability to impact cloud ice formation processes and may originate from a range of local to long-range transported sources. As a result, several INP quantification and characterization studies have been conducted in clouds at mountaintop atmospheric research facilities, such as Storm Peak Laboratory in the United States (Baustian et al., 2012; Cziczo et al., 2004; Richardson et al., 2007), Puy de Dôme in France

(Joly et al., 2014; Joly et al., 2013), and Jungfraujoch in Switzerland (Chou et al., 2011; Conen et al., 2015; Stopelli et al., 2017; Stopelli et al., 2016). Such studies provide routine or long-term measurements of INPs in clouds, yet one disadvantage is that profiling is not possible. Vertical profiling of INPs can serve as a connection between the ground and various altitudes below, in, and above cloud. Targeted aircraft campaigns have helped explain the role of INPs in cloud ice formation at all levels from below cloud, cloud base, in-cloud, and cloud top (e.g., Avramov et al., 2011; Creamean et al., 2013; Curry et al.,

2000; DeMott et al., 2010; DeMott et al., 2003; Pratt et al., 2009; Prenni et al., 2009a; Rogers et al., 2001; Rogers et al., 1998; Schnell, 1982). Although such campaigns yield results crucial for understanding the vertical distribution of INPs in cloudy environments, they are intensive with regard to personnel, cost, and time.

Overall, a key gap in ice nucleation research is routine vertical profiling of INP abundance, efficiency, and chemical and physical characterization (Coluzza et al., 2017). Tropospheric measurements via balloon-based systems have been a desirable

means of measuring aerosol properties on an inexpensive and thus, more frequent basis. However, such measurements can be limited in terms of time, measurements made, or location. For example, long-term records of tropospheric aerosol particle size distributions have been reported in Wyoming, United States (i.e., 20 years) (Hofmann, 1993). The same launched balloon system was deployed in Antarctica, demonstrating the utility of this platform in multiple environments (Hofmann et al., 1989). Particle size distributions have also been measured via launched balloons in several locations in China using optical particle

counters (Iwasaka et al., 2003; Kim et al., 2003; Tobo et al., 2007). One major caveat with these studies is that it is not clear if the balloon systems were retrievable, given their maximum flight ceilings were located well into the stratosphere. In addition, the launched balloon platforms provide information on 1 – 2 aerosol profiles (i.e., ascent and sometimes descent) and are limited by payload weight. Particle spectrometers have also been deployed and retrieved on tethered balloon systems (de Boer et al., 2018; Greenberg et al., 2009; Maletto et al., 2003; Renard et al., 2016; Siebert et al., 2004; Wehner et al., 2007), affording

information on aerosol layer locations and evolution by means of multiple profiles. A few studies have deployed miniature aerosol filter samplers on launched or tethered balloon systems, yielding information on aerosol chemistry (Hara et al., 2011; Rankin and Wolff, 2002).; however, such samplers contained one filter per flight, thus providing information on aerosol

properties at only one altitude (i.e., not a profile). A noteworthy study by Ardon-Dryer et al. (2011) consisted of measurements of immersion mode INP concentrations from a tethered balloon flight in Antarctica, although only at temperatures below −18 °C from three filters collected below 200 m above ground level (a.g.l.). In general, tethered balloons can handle much larger payloads than launched systems, but are limited to lower altitudes (i.e., up to approximately 2 km a.g.l. anywhere), have wind condition limitations, and involve more complicated logistics (e.g., use of a winch and personnel required to operate a winch) thus may not be ideal for sampling INPs in all conditions. Schrod et al. (2017) present INP measurements from several flights using unmanned aircraft systems (UASs) over the Eastern Mediterranean, but only in the deposition nucleation mode. To our knowledge, the results from Ardon-Dryer et al. (2011) and Schrod et al. (2017) are the only reported vertical INP measurements using smaller, unmanned systems. The fact that only two published studies exist, in addition to the limitations of such studies (and our limitations as discussed in more detail herein) demonstrate the challenges associated with obtaining INP measurements aloft.

Overall, both launched and tethered balloon platforms, and UASs, have their advantages and disadvantages in terms of flight ceiling, profiling, retrieveability, cost, operational logistics, and payload restrictions. A solution to reduce the limitations of these methods is a launched balloon system that can be controlled in terms of ascent and descent, affords multiple profiling and payload retrieval capabilities, and collects aerosol loadings sufficient for altitude-resolved offline ice nucleation measurements. Here, we present a measurement system called HOVERCAT (Honing On VERtical Cloud and Aerosol properTies) deployed on an experimental launched balloon system that possesses such capabilities.

## 2 Methods

The first prototype of HOVERCAT was recently built and tested in Colorado, United States, consisting of an aerosol module for measuring real-time particle size distributions and a miniaturized filter sampler for aerosol collection for offline ice nucleation analyses. The balloon platform, called the Boomerang Balloon Flight Control System (BBFCS), was used to fly HOVERCAT. The current version of HOVERCAT is experimental, thus we consider it as in Phase I of its development, and is described herein. As discussed later, we provide future directions for modification and improvement of HOVERCAT and recommendations for non-tethered balloon systems in general for future deployments.

### 2.1 HOVERCAT: The aerosol instrumentation package

The aerosol module package contains: 1) an optical particle counter (Alphasense OPC-N2) for particle size distributions (16 size bins for 0.38 – 17 μm in diameter) and estimated particle mass concentrations with optical diameters ≤ 1, 2.5, and 10 μm ($PM_1$, $PM_{2.5}$, and $PM_{10}$, respectively) and 2) the National Oceanic and Atmospheric Administration (NOAA)-built miniaturized Time-Resolved Aerosol Particle Sampler (TRAPS) for collection of up to 10 samples. The time resolution can be set at the desired rate but was set for 30 minutes in the current study. The OPC-N2 operates at 175 mA in operation mode and weighs

105 g. Flow rates are adjusted based on ambient pressure to maintain a 1.2 L min$^{-1}$ flow using a patented 'pump-less' design. Data are stored on a microprocessor within the OPC during collection. A default density of 1.65 g mL$^{-1}$ and refractive index of 1.5 were used to estimate particle mass concentrations. The TRAPS design is based on the filter components of the NOAA Continuous Light Absorption Photometer (CLAP), without the optical components and measurements (Ogren et al., 2017)

(Figure 1a). It is connected to a small 12 V DC vacuum pump (Brailsford & Co., Inc. TD-4X2N), which nominally enables a flow rate of approximately 1.2±0.1 L min$^{-1}$ through the TRAPS when a 47-mm diameter filter with 0.2 µm pore size is in place. A Honeywell AWM43600V mass flow meter measures sample flow rate. Ten miniature solenoid valves select the active sample spot and are controlled by an on-board microprocessor preselected for the desired time resolution, which was 30 minutes per sample spot for the HOVERCAT test flights. The TRAPS flow rate at 30 minutes provides approximately 40 total

litres of air through each spot, which is ideal for measuring more realistic INP concentrations (Mossop and Thorndike, 1966). Sample loaded spots average to a coverage area of 19.9 mm$^2$ (equates to a spot diameter of approximately 4.46 mm). The TRAPS has the highest collection efficiency for particles in the 1 nm – 10 µm aerodynamic diameter range—with particle losses of less than 10% for 5 nm – 7 µm particles and less than 1% for 30 nm – 2.5 µm particles at 1.0 L min$^{-1}$—but can collect particles with larger diameters (Ogren et al., 2017).

The TRAPS, micropump, and OPC are all operated by battery: the TRAPS and micropump run off a battery pack containing three 18650 rechargeable Li-ion batteries (Panasonic NCR18650B, 12 V output, 3400 mAh) and the OPC runs off one rechargeable battery (Anker PowerCore 5000, 5 V output, 5000 mAh). The OPC can operate for several days on its portable battery, while the TRAPS and pump can operate for up to 5 hours on its battery pack. Both the TRAPS and OPC are connected to inlets composed of an 8-inch segment of ¼-inch ID black conductive tubing connected to a stainless-steel funnel (5 cm in

diameter) with the opening covered with stainless steel mesh. All components are seated in a foam enclosure with removable lid and inlets extending out of the bottom (Figure 1b).

## 2.2 BBFCS: The balloon platform

The BBFCS is a real-time, remote device that allows the user to control the ascent and descent of standard latex weather balloons (Figure 1c). The primary features are a lift-gas vent valve in the control module that permits negative buoyancy

adjustments and a sand ballaster (i.e., ballast module) that permits positive buoyancy adjustments. Buoyancy adjustments as small as 5 g of lift are possible. For example, if a faster or slower ascent is desired, ballast can be dropped or venting can be done, respectively. If descent is desired, a longer and faster venting is applied. Due to the ability to slow down the fall speed by a combination of the appropriate amount of venting and dropping ballast, if needed, landing the system is relatively gentle and did not result in instrumental damage during the test flights. It is possible the balloon itself can be reused (i.e., we used the

same balloon for two flights).

Two-way communication is achieved through a 70-cm line-of-sight LoRa radio link. The system features a 1/4W transceiver that uses a low baud rate and a slow 4-second time-division multiple access (TDMA) cycle to achieve ranges in excess of 300 km. The system also features redundant termination methods, anti-collision strobes, positioning, and flight sensors. A recovery parachute is included for emergency termination and faster fall speeds than slow balloon deflation. The BBFCS was manually controlled for this project. We utilized a software interface on a ground-based computer to analyse the real-time flight conditions and send the necessary buoyancy control commands to achieve the desired flight profile. We drove in the approximate trajectory of the balloon in order to stay within the 300-km communications range, thus were able to physically retrieve it when it ultimately landed. Early morning launches were conducted to maximize the calm low-troposphere atmospheric conditions as flight control is much easier in such conditions. Because this project entailed low-altitude flights that did not exceed 9.6 km above mean sea level (a.m.s.l.) or approximately 8.1 km a.g.l., 300 g latex balloons were used. These relatively small balloons, for a 3.9-kg payload, ensured that the envelope was always under tension and would expel lift-gas whenever the vent valve was opened, while ensuring that the burst altitude was above the expected operational altitude. Burst altitude was calculated to be 13 – 14 km a.m.s.l. (11.5 – 12.5 km a.g.l.) depending on how much lift gas had been vented. The BBFCS is designed to allow Federal Aviation Administration (FAA) part 101 exempt flights, even when carrying a reasonably-sized payload (i.e., total payload weight of less than 5.5 kg and no one module greater than 2.7 kg).

## 2.4 Test flight details

The overall launch mass was 4250 g with 450 g of free lift to achieve an initial 3 m s$^{-1}$ ascent rate. System masses were: 350 g for the balloon and connection spindle, 900 g for the control module and parachute, 2300 g for HOVERCAT, and 700 g of ballast. Initial flight planning called for a 5-step flight profile with 500-m altitude steps. This allocated 100 g of ballast per step, 1.5 m s$^{-1}$ anticipated ascent rate between steps, with a 200-g reserve for the flight to help maintain the desired altitude. However, this plan was ultimately not executed due to flight complications discussed herein. The flight train for this project consisted, from top to bottom: latex balloon, valve and flight computer modules, 500 mm of line, aerosol module, 500 mm of line, and ballast module (Figure 1d). The recovery parachute was attached to the bottom of the flight computer module and hung off to the side. The parachute's apex was attached to the termination clamp and was released by this clamp during termination or by aerodynamic drag if the balloon had prematurely burst. The OPC was started during balloon inflation and the TRAPS and micropump were started via Bluetooth just prior to take off. Two miniature cameras (Mobius Basic ActionCam with wide angle lens) were mounted to and facing the BBFCS valve module and HOVERCAT for time lapse photos during take-off, flight, and landing.

Three test flights were conducted in central Colorado during 24 – 26 May 2017. Two of the three flights had instrument operational issues (i.e., 24 and 26 May), so only data from the 25 May flight is presented herein. Briefly, communications were lost during the 24 May flight and as a result, controlling the valve and ballast modules was not possible. The system reached 8.1 km  a.g.l. and ambient pressure was too low for the TRAPS pump to operate. The 26 May flight reached > 2 km

a.m.s.l. (> 500 m a.g.l.), in which the TRAPS pump also did not operate correctly. For both the 24 and 26 May flights, the total volume of air pulled through the filters was 1 – 12 L above 2.5 km a.m.s.l. (1.1 km ), equating to loadings too low for offline analyses (i.e., calculated INP concentrations were below detection limits). Based on the successful 25 May flight and unsuccessful flights on 24 and 26 May, we have concluded that in its current configuration, HOVERCAT can operate below 2.5 km a.m.s.l., otherwise at the low pressures, the current micropump cannot generate sufficient flow. New, higher volume pumps are being tested.

The 3-dimensional flight path for 25 May is shown in Figure 2. The horizontal distance between launch and landing was 16.8 km, bird's eye view. Conditions were partly cloudy with surface air temperatures ranging from 16 – 21 °C, relative humidity from 35 – 47%, and wind speeds from 2 – 3 m s$^{-1}$ from the north and south (hourly meteorological data during flight times obtained from the Colorado Department of Public Health and Environment (CDPHE) at the Boulder Reservoir ground site; 40.07°N, 105.22°W; https://www.colorado.gov/pacific/cdphe/data). HOVERCAT did not fly through the clouds present that day, but remained below cloud base, based on visual identification of the system while tracking in real-time (i.e., the system was always in line-of-sight).

**2.3 Offline ice nucleation analyses**

**2.3.1 Drop freezing assay for immersion mode ice nucleation**

For the 25 May flight, aerosol samples were collected on 47 mm filters (Pallflex® EmFab™). Pre-treatment of the filters by means of a 6 N nitric acid bath (Certified ACS Plus, Fisher Scientific), 3 times rinse with ultrapure water (UPW; Barnstead™ Smart2Pure™ 6 UV/UF), and baking at 150 °C for 30 minutes, was conducted to remove possible filter INP artefacts. Out of the filters tested, EmFab™ possessed the lowest contribution from artefacts compared to cellulose nitrate and polytetrafluoroethylene and survived the pre-treatment process.

Immersion mode freezing was tested using a drop freezing assay (DFA) cold plate apparatus. This cold plate technique was based on previous but slightly modified apparatuses (Hill et al., 2016; Stopelli et al., 2014; Tobo, 2016; Wright and Petters, 2013). For brevity, we call this system the NOAA Drop Freezing Cold Plate (DFCP). Following collection and prior to analysis, sample filters were stored frozen for approximately six months. After removing from the freezer, each sample spot was carefully cut and separated from the 25 May filter; six spots (i.e., samples) were successfully collected before the battery died. Each spot was placed is a 29-mL sterile Whirlpak® bag with 2 mL of UPW to resuspend particles deposited on the filter. The bags were sealed and shaken at 500 rpm for two hours (Bowers et al., 2009). Copper discs (76 mm in diameter, 3.2 mm thick) were prepared by cleaning with isopropanol (99.5% ACS Grade, LabChem. Inc.), then coated with a thin layer of petrolatum (100%, Vaseline®) (Bowers et al., 2009; Tobo, 2016). Three of the spots on the filter had visible aerosol deposits that were successfully transferred to the UPW (i.e., based on visual identification).

Following sample preparation, a sterile, single-use syringe was used to draw 0.25 mL of the suspension and 100 drops were pipetted onto the petrolatum-coated copper disc, creating an array of ~2.5-μL aliquots. Drops were visually inspected for size; however, it is possible not all drops were the same exact volume, which could lead to a small level of indeterminable uncertainty. However, previous studies have elucidated that drops need to be orders of magnitude different in volume to significantly perturb the freezing temperature from drop size, alone (Bigg, 1953; Hader et al., 2014; Langham and Mason, 1958). The copper disc was then placed on a thermoelectric cold plate (Aldrich®) and covered with a transparent plastic dome. Small holes in the side of the dome and copper disc permitted placement of up to four temperature probes using an Omega™ thermometer/data logger (RDXL4SD). The Omega™ meter has a 0.1 °C resolution and accuracy of ± (0.4% + 0.5 ℃) for the K sensor types used. During the test, the cold plate was cooled at $1 - 10$ °C min$^{-1}$ from room temperature until all drops on the plate were frozen or until the DFCP detection limit of approximately −32 to −33 °C. Control experiments with UPW at various cooling rates within this range show no discernible dependency of drop freezing on cooling rate, akin to previous works (Vali and Stansbury, 1966; Wright and Petters, 2013). Frozen drops were detected visually, but recorded through software written in-house, providing the freezing temperature and cooling rate of each drop frozen. For the control experiments with UPW, some experiments resulted in unfrozen drops at the DFCP lower temperature limit, thus, the fraction frozen was calculated from the number of drops detected, including the unfrozen remaining, which is the reason why not all fractions frozen = 1. However, all drops froze for tests with blanks for the sample handling and the samples themselves. Each sample was tested three times with 100 new drops for each test. From each test, the fraction frozen and percentage of detected frozen drops were calculated. The results from the triplicate tests are then binned every 0.5 °C to produce one spectrum per sample.

Although the methodology behind DFA is well established, control experiments were conducted with UPW for full system characterization of the DFCP. First, temperature differences were measured within the range of cooling rates using UPW on petrolatum-coated copper discs between the centre of the disc (thermocouple inserted in a small diameter hole in the side of the disk) and a drop on top of the plate with a thermocouple inserted into the drop (Figure 3). As expected based on previous work (Vali and Stansbury, 1966; Wright and Petters, 2013), there was no dependence of the temperature difference on cooling rate, but on average, the drop temperature was 0.33±0.15 °C warmer than the centre of the plate. Thus, a +0.33 °C correction factor was added to any temperature herein and an uncertainty of 0.15 °C was added to the probe accuracy uncertainty.

Second, various hydrophobic coatings with UPW were tested for the best combination of materials to use with the least influence from artefacts (Figure 4). Materials tested were chosen based on those used in previous work and included: 1) direct petrolatum (Tobo, 2016), 2) 15% w/v petrolatum in xylenes (Certified ACS Reagent Grade, Ricca Chemical) (Bowers et al., 2009), 3) silicone fluid (710 fluid, Dow Corning®) (Polen et al., 2016), and 4) squalene (≥ 98%, Sigma-Aldrich®) (Hader et al., 2014; Wright and Petters, 2013; Wright et al., 2013). The silicone fluid was difficult to use for cold plate experimentation because droplets would coalesce during the experiment and freezing detection by eye was difficult due to the glare of the substance. Squalene was less viscous than the silicone fluid, inducing more drop coalescence but freezing detection was easier

than the silicone fluid. Both materials remained in the fluid state, thus are not ideal for direct cold plate use, but have been proven suitable for cold stages that use covered sample dishes or trays and smaller drop sizes (Hader et al., 2014; Polen et al., 2016; Wright and Petters, 2013; Wright et al., 2013). The petrolatum and xylenes solution creates a thin layer of petrolatum after drying to evaporate the xylenes and alleviate the coalescence problem; however, as evidenced by the freezing spectra in

Figure 4, is not the best option in terms of limiting artefacts. To summarize, a hydrophobic coating is needed on the copper plate and the option with the least influence from contaminants is direct petrolatum smeared onto the plate using UPW.

Last, the effect of drop size was tested using UPW and petrolatum-coated copper plates (Figure 5). Normally, 2.5-µL drops are created by hand using a sterile syringe. Because such drops are created without the use of a pipette, possible small variations in drop volume may occur. The same volume drops were created with a pipette and sterile tips and tested against syringe drops.

Additionally, tests with 1.5-µL and 5.0-µL drops were conducted to evaluate the effects of larger changes in volume. One major caveat with the pipette technique is that it takes substantially more time to create the arrays of 100 drops (approximately five times slower than the syringe method). Overall, the best method in terms of onset freezing temperatures and fraction frozen was the 2.5-µL drops created via syringe. This test was comparable in terms of fraction frozen to the 1.5-µL drops colder than −21 °C. One possible explanation for the higher onset temperature and higher concentrations of impurities in the

2.5-µL pipetted drops as compared to the 2.5-µL syringed drops is contamination from the pipette tips. The 5.0-µL test demonstrated that drops of this size are too large such that they induce freezing at warmer temperatures and are subject to large variability—in theory, the larger the drop volume, the larger the abundance of impurities within a single drop that may facilitate ice formation (Bigg, 1953). Overall, our drop size tests demonstrate the efficiency and reliability of 2.5-µL drops created via syringe.

Out of the 100 drops for each test, 95±5% on average (ranging between 84 – 100%) were detected as frozen and recorded from all tests (Figure 6). Some of the tests within the same sample were reproducible within error, demonstrating the reliability of the method (e.g., samples 1 and 3). However, variability from test-to-test within the same sample could occur due to: 1) detection of rarer INPs at specific temperatures during 1 – 2 of the tests or 2) uncertainties arising from instrumental artefacts, such as contamination between tests. These results demonstrate the importance of running triplicate (or more) tests for DFA

techniques—to capture some of the rarer INPs that may exist in the samples or account for test-to-test variabilities. Such rarer INPs may be missed or over accounted for if only one test is conducted. The cooling rate was variable during each test, but maintained within the 1 – 10 °C min$^{-1}$ range and the fraction frozen did not show a noticeable dependence on the cooling rate, as discussed above.

From the fraction of drops frozen and the known total volume of air per sample, we calculated the estimated INP concentration

(L$^{-1}$ of air) with the universally applied equation by Vali (1971):

$$[INP](L^{-1}) = \frac{\ln(1-f)}{V_{drop}} \times \frac{V_{suspension}}{V_{air}}$$

where $f$ is the proportion of droplets frozen, $V_{drop}$ is the volume of each drop, $V_{suspension}$ is the volume of the suspension (i.e.,
2.5 mL for the sample tests), and $V_{air}$ is the volume of air per sample. We averaged the total volume of air from the six field
samples collected and applied that to the equation to calculate INP concentrations for the blanks, in order to conduct a direct
comparison and evaluate the INP concentrations in the samples relative to the blanks.

### 2.3.2 Raman microscopy for deposition mode ice nucleation

Depositional ice nucleation was measured using a Nicolet Almega XR Dispersive Raman Spectrometer outfitted with a
Linkham THMS600 environmental cell and a Buck Research CR-1A chilled-mirror hygrometer. The Raman spectrometer was
coupled with an Olympus BX51 research-grade optical microscope with 10x, 20x, and 50x magnification abilities. The
environmental cell and CR-1A hygrometer allow for temperature control and dew/frost point measurements to back calculate
saturation ice ratios, $S_{ice}$. The environmental cell was connected to two UHP grade $N_2$ tanks, one is humidified and the other
is a "dry" tank that is not humidified. These two were then mixed, fed through the environmental cell, and lastly the CR-1A
measures the dew/frost point. In these experiments, the water vapour was kept constant while the temperature was decreased,
which resulted in an increase in $S_{ice}$. This experimental set up, calibration, and calculation is explained in more in detail in
Baustian et al. (2010), Schill and Tolbert (2013), and Primm et al. (2017).

An aliquot of the solutions from the previous immersion mode experiments were used for deposition mode ice nucleation
experiments (i.e., untested sample solution). The solution derived from each spot on the collected filter sample was nebulized
onto a fused silica disc, which was then placed into the environmental cell at ~0% RH to allow for evaporation of water from
the particles. The temperature was then decreased at a rate of 0.1 K min$^{-1}$, while water vapour was held constant. Temperature
and dew point were recorded during the entire experiment. $S_{ice}$ was determined from the temperature and dew point where ice
was first visually identified. The different $S_{ice}$ values at different temperatures were determined by performing the same
procedure, but changing the starting water vapour pressure. This difference in water vapour pressure changes the $S_{ice}$ value at
different temperatures. Temperatures which were analysed for depositional ice nucleation were chosen to cover a wide range
of those previously reported and relevant for several cloud regimes (Hoose and Möhler, 2012). Nebulization onto the disc
resulted in 5000 – 10000 particles, with a range of 1 µm to 50 µm in diameter, deposited on the surface depending on the spot
from the filter paper. Of the particles that nucleated ice, 3 – 5 particles were analysed for composition using Raman
spectrometry for each sample. Because the purpose of the analysis was to prove that particles could be analysed for depositional
ice nucleation using samples collected by HOVERCAT, only the first few particles that formed ice at each temperature regime
were recorded. A more statistical approach (i.e., analysing more particles) to characterize the depositional INP population
during the flight is outside the scope of this manuscript.

## 3 Results and discussion

### 3.1 Operation of HOVERCAT instruments during test flight

Although the ability to control the exact altitude of the system was difficult due to vertical winds—which was determined by abrupt ascent or descent and horizontal transport while tracking in real-time—we were able to control gas venting and dropping ballast to slow down ascent and descent, and sample at altitudes from the ground level up to 2543 m a.m.s.l. (approximately 1053 m a.g.l.) for 3 hours (Figure 7). The ability to control the BBFCS to execute the step-wise flight plan was difficult given the winds and the several-second delay in time when venting or dropping ballast to decrease or increase in altitude, respectively. Minor fluctuations in BBFCS control to maintain altitude was not possible during 25 May conditions, but may be on a calmer day aloft. Because of such issues, the first two profiles (i.e., ascent followed by descent to ground) during the first hour of flight (up to 2316 and 2543 m a.m.s.l.) were abrupt and parking at desired altitudes was not achieved. We were able to maintain altitude at 1771±80 m a.m.s.l. (281 m a.g.l.) during the third profile (08:00 – 09:00), with a short drop in altitude around 08:50. Starting at 09:07, we were able to maintain altitude just above the ground at 1536±20 m a.m.s.l. (46 m a.g.l.) until 9:15, with a final profile up to 2098 m a.m.s.l. (608 m a.g.l.) at 09:25. Ultimately, the balloon deflated and ended the flight at 09:36.

While controlling the exact altitude of the BBFCS was difficult, the aerosol measurements were fruitful. The OPC measured particle concentrations up to 250 cm$^{-3}$ while at the ground (average of 6 cm$^{-3}$), with the lowest concentrations occurring at the highest altitudes (< 1 to 2 cm$^{-3}$; average of 1 cm$^{-3}$). However, episodic spikes in number occurred when stable at the ground, indicating localized sources of high concentrations of particles. PM concentrations followed a similar inverse relationship with altitude (Figure 7). The total flow though the filter in TRAPS was fairly consistent throughout the flight, starting at 40 L for Sample 1 and decreasing to 32 L for Sample 6. The slight decrease possibly resulted from: 1) inconsistent power supply by the battery pack to the micropump or 2) strain on the micropump with altitude, although the latter is less likely given the variability in altitude throughout the flight.

### 3.2 Immersion freezing ice nucleation

From the six filter sample spots that were collected, aerosol loading was sufficient to conduct INP measurements using the DFCP system. Cumulative INP spectra show relatively low concentrations (i.e., $10^{-2} – 10^{-1}$ L$^{-1}$) of warm temperature INPs (> −10 °C, likely of biological origin (Murray et al., 2012)) for all samples, while reaching up to $10^{1}$ L$^{-1}$ at temperatures below −20 °C (Figure 8). Such concentrations are within range of those previously reported in Colorado: Prenni et al. (2013) reported $1 – 10^{2}$ L$^{-1}$ at −25 °C. The highest INP concentrations were observed from Sample 3, which corresponded to the time where HOVERCAT was closest to the ground (i.e., 69% of sample time was < 50 m a.g.l.), on average (Figure 9a). Sample 6 had the highest concentrations of INPs active between −8 and −12.5 °C, which also corresponds to when HOVERCAT hovered just above ground level (19% of the time; Figure 7). It is important to note that all samples aside from Sample 4 hovered near the

ground: Samples 1, 2, and 5 were close to the ground 40%, 9%, and 2% of the time, respectively. Thus, altitude-dependent results could be skewed by collection nearest to the local source of aerosol. It is important to note that the samples that spend little to no time at the ground corresponded to the lowest INP concentrations (i.e., Samples 4 and 5). However, based on OPC number concentrations, there was not always a clear decrease of aerosol concentrations with altitude (e.g., Sample 5).

Additionally, concentrations were calculated and based on total volume of air, indicating that the altitude in which the sample was collected at for the most amount of time is representative of the overall sample INP population. Combined, the immersion INP, OPC, and BBFCS results indicate that: 1) total particle number concentrations and INP concentrations were highest when HOVERCAT sampled near the ground and 2) INPs of likely biological origin remained close to the surface, which is predominantly agricultural soils in this region (Hill et al., 2016). The relative abundance of INPs to total particles is also

consistent with previously reported values (DeMott et al., 2010): INPs represented 1 in every $10^2$ to $10^5$ number of particles detected by the OPC, although the OPC does not measure below 380 nm so the fractions might in reality be even lower (Figure 9b). However, INPs are thought to be relatively large (i.e., > 200 nm in diameter) based on previous work (DeMott et al., 2010; Fridlind et al., 2012; Kanji et al., 2017; Mertes et al., 2007; Niedermeier et al., 2015), so the OPC may be relevant for supporting INP measurements. Although these results may not be surprising (e.g., total particle, INP concentrations within range of

previous work and generally highest near the ground, and biological INPs sourced from an agricultural region) and yield results consistent with previous work (DeMott et al., 2010; Hill et al., 2016; Murray et al., 2012; Prenni et al., 2013), they demonstrate the utility and reliability of the collection and analytical methods of HOVERCAT and the DFCP systems.

### 3.3 Deposition ice nucleation

Depositional ice nucleation analysis of the six filter samples was conducted using the extra volume of resuspension left from

the immersion freezing analysis (i.e., the portion of the 2 mL that was not used on the DFCP). Of the particles that nucleated ice, 3 – 5 particles were analysed for composition using Raman spectrometry for each sample. We assume that a majority of the particles are of similar concentration because the whole sample was dissolved in water, allowed to mix to a homogeneous solution, and nebulized onto the sample disc. Indeed, the particle composition was similar for each particle in any sample, while there was variation from sample to sample. Although the Raman spectral and ice nucleation analyses are helpful to

observe the overall particle composition as temperature and relative humidity are changed, the experiment does not determine the size or mixing state of the particles as they were in the atmosphere. Further, the spectral resolution of 1 micrometre in our system does not allow smaller scales to be distinguished within the individual particles probed.

Overall, ice activation onset conditions between the six samples were similar at all temperatures tested (Figure 10). However, at –40 °C, Samples 3 and 4 showed first ice nucleation activity at a saturation ice ratio of 1.12, which were lower than the

other samples and may be characterized as more efficient deposition INPs at that temperature as compared to the remaining samples These samples contained slightly more efficient INPs at −25 °C, but similar efficiencies to the remaining samples at −55 °C. Raman spectrometry demonstrates that most of the samples were compositionally disparate from each other (Figure

11). The first three samples show a very intense fluorescence signal (i.e., the curve-like characteristic of the baseline), which is consistent with either biological or organic materials (Baustian et al., 2012). Additionally, Sample 2 contained a peak for carbonate, which is indicative of a mineral dust signature (Baustian et al., 2012). The sample collection time periods for these samples occurred directly over a dense agricultural region in the Colorado plains, supporting the observation of highly fluorescent particles (Figures 2 and 7). Interestingly, Sample 3 contained efficient immersion mode INPs as well, that were likely of biological origin due to the relatively higher INP concentrations at temperatures greater than −10 °C (Figure 8). Samples 4, 5, and 6 show a C-H stretch peak, and occasionally sulphate ($SO_4^{2-}$) and nitrate ($NO_3^-$) peaks, which is consistent with the composition of typical anthropogenic aerosols in the atmosphere (Zhang et al., 2007). Sample 5 had the most intense anthropogenic peaks while yielding the least efficient immersion mode and deposition mode (i.e., at the two highest temperatures measured) INPs. It is possible any INPs present in this sample were affected by sulphate or nitrate coatings, which have been shown to inhibit the ice nucleating abilities of aerosols (e.g., Cziczo et al., 2009; Möhler et al., 2008; Reitz et al., 2011; Sullivan et al., 2010). Collection of Samples 5 and 6 coincided with when HOVERCAT flew close to the ground near I-25, where vehicular traffic and industry lining the multilane interstate likely contributed to the larger signal from anthropogenic functional groups and less efficient INPs. However, the Raman spectrum for Sample 6 also has a weak fluorescent signature, indicating a possible biological contribution. HOVERCAT flew from over I-25 to the west over more agricultural lands. Sample 6 also contained high concentrations of INPs at –10 °C, indicating the sample also contained biological INPs. Combined, these results from Sample 6 suggest a mixture of biological and anthropogenic sources.

### 3.4 Recommendations for future airborne INP measurements on small platforms

As indicated earlier, Phase I of the BBFCS and HOVERCAT combination exists in its current prototype state. The priorities of Phase I were to develop a system that is cost effective, user-friendly, versatile, and in compliance with FAA regulations without the need for special approvals or restricted airspace. Under these priorities, our objectives were to address if we could develop such a system that was: (1) recoverable and (2) controllable. Recoverability was a requirement as we needed to obtain the filter samples for the offline INP analysis, while controllability was an added benefit to have altitude-resolved INP measurements. We successfully achieved the first objective by recovering the system after it landed and controlling the BBFCS such that the landing was not damaging to the instrumentation. The second objective, however, is still in need of improvement as discussed here. The benefits of the system as a whole are that it is cost effective and easy to operate relative to traditional airborne measurements of INPs, and did not require special FAA approvals, providing flexibility to fly anywhere at any time. HOVERCAT alone has the benefit of having time-resolved filter sampling capabilities that, if able to control altitude, would yield vertically-resolved INP measurements. However, as discussed throughout, both the BBFCS and HOVERCAT have their limitations. Here, we discuss these limitations and provide recommendations not only for a Phase II system for HOVERCAT, but also recommendations generally applicable towards INP measurements on small airborne platforms.

First, HOVERCAT could only operate in its current design up to 2.5 m a.m.s.l. (1.1 m a.g.l.). Although this is an improvement over previously reported tethered measurements of INPs (e.g., Ardon-Dryer et al. (2011) reached 196 m a.g.l.), achieving higher altitudes is desired to capture the profile of INPs in and above clouds using a launched platform that affords the flexibility to essentially fly anywhere. To improve operation for higher altitudes, modifications should be made to incorporate a stronger micropump that would yield higher flows and operation at lower pressures. The main issue is that to fly at free will (i.e., under FAA compliance), payload weight must be maintained under 2.7 kg for any single module (i.e., HOVERCAT). Thus, stronger pumps, which are by nature heavier, may not be realistic for HOVERCAT on a launched balloon system. Implementing a stronger pump would require either: (1) a FAA Certificates of Waiver or Authorization (COA), (2) flights in restricted airspace, or (3) flights on a tethered balloon system, all of which do not align with the priorities to maintain simplicity and versatility. However, we generally recommend future parallel measurements be made with a better pump. One option could be to reduce weight of the other components (e.g., replace the metal protective enclosure of the TRAPS with lightweight foam). This may not afford enough margin to incorporate the weight of a better pump, but is a possible alternative that needs to be tested.

Second, the hovering capability needs improvement, either by further testing with the BBFCS or modification to a traditional launched balloon system. We were able to control the altitude ±80 m, but executing the step-wise flight plan proved to be more difficult than anticipated. The venting and ballasting functioned properly, but room for improvement could be focused on accounting for natural conditions (i.e., updrafts and downdrafts) that affect the altitude and truly enable the BBFCS to hover at desired altitudes. As another option, HOVERCAT could be deployed on a traditional launched balloon with a slow rise rate and less helium, or a reverse parachute (i.e., less buoyancy and more drag) to afford a steady vertical profile, although this eliminates the hovering capability of the system unless the free lift is adjusted such that the system may hover near inversions For instance, an ascent rate of 0.5 m s$^{-1}$ would provide a 900 m vertical resolution (at 30 minutes per sample). If such a system were successful, the need for bidirectional communication to control TRAPS sampling intervals would not be required and would eliminate the need for additional hardware, receivers, batteries, and other data processing components in HOVERCAT and for the ground station. In general, we recommend implementing advanced controllability features into traditional launched balloons to not necessarily hover, but afford a consistent and slow rise for sample collection, and components to terminate the flight at the desired altitude such that the package is still recoverable.

Third, the Phase I pilot study involved sampling in clear air to conceptually prove HOVERCAT could perform as desired. Ideally, operation of such a system would be in clouds and harsher conditions such as the Arctic. To function in harsher environments, testing the modules in humidified, pressure-controlled, and temperature-controlled conditions is required at temperatures down to −40 °C. Ardon-Dryer et al. (2011) measured INPs successfully using a filter sampler in the Antarctic, but did not collect samples in cloud. Schrod et al. (2017) deployed their sampler on a small unmanned aircraft system up to 2.5 km a.g.l., but did not fly in cloud or ambient temperatures below approximately 15 °C. Combined, even though our system

and these previous systems are subject to limitations, they are a significant advancement towards a more flexible and versatile manner in which INPs above ground level can be measured. In general, additional research is needed to continue to improve such systems with regard to cost, performance, and enhanced spatial and temporal coverage to improve understanding of INP impacts on clouds.

## 4 Conclusions

Here, we present a novel airborne aerosol and ice nucleation measurement system called HOVERCAT that was tested during a pilot study on the BBFCS platform. HOVERCAT measured time-resolved particle number and INP concentrations a range of altitudes up to 2.6 km a.m.s.l. (1.1 km a.g.l. at a ground elevation of 1.5 km a.m.s.l.). Although controlling the ascent and descent of the balloon platform was difficult, we provide recommendations for future platforms and measurements using similar non-tethered balloon systems. Unlike similar systems, HOVERCAT can vertically resolve particle number concentrations in addition to both immersion and deposition mode INPs. To our knowledge, this is the first platform to perform such measurements in tandem. Phase I of HOVERCAT has been tested, while ongoing efforts for improvement and modification are desired for Phase II to enable HOVERCAT to fly higher and in more inclement conditions.

The ability to evaluate vertical distributions of INP concentrations and glaciation temperatures is of crucial importance in order to inform and constrain process level models to improve understanding of aerosol-cloud interactions. Additionally, more routine measurements of INP properties are needed to understand the evolving nature of aerosol-cloud interactions under a wide range of cloud regimes, locations, and time of year.

### Acknowledgements

We would like to acknowledge the Cooperative Institute for Research in Environmental Sciences at the University of Colorado, Boulder, USA for funding this work. Also, the equipment for offline drop freezing analysis was purchased using funding from the NOAA Climate Program Office (CPO).

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

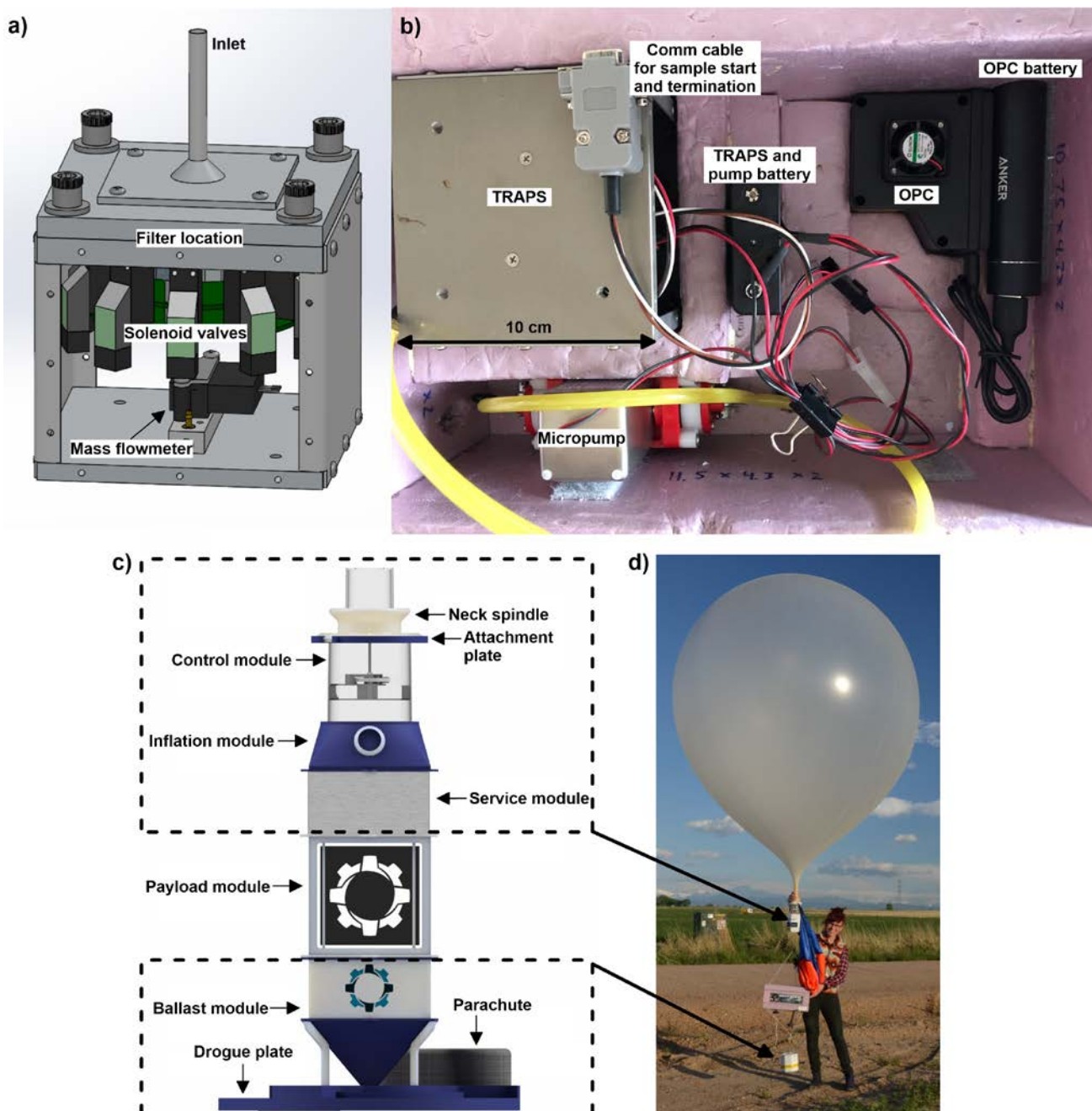

**Figure 1: Components of the complete flight system, including a) schematic of the TRAPS, b) picture of the aerosol module, c) schematic of the BBFCS, and d) flight train for test flights. Note that the service module on the BBFCS was separated approximately 1 m from the ballast module with the aerosol module (i.e., payload) in between. The ballast module was controlled by the on-board computer in the control module via an extended cable that ran down the tether string. The separated BBFCS modules were housed in foam for flights.**

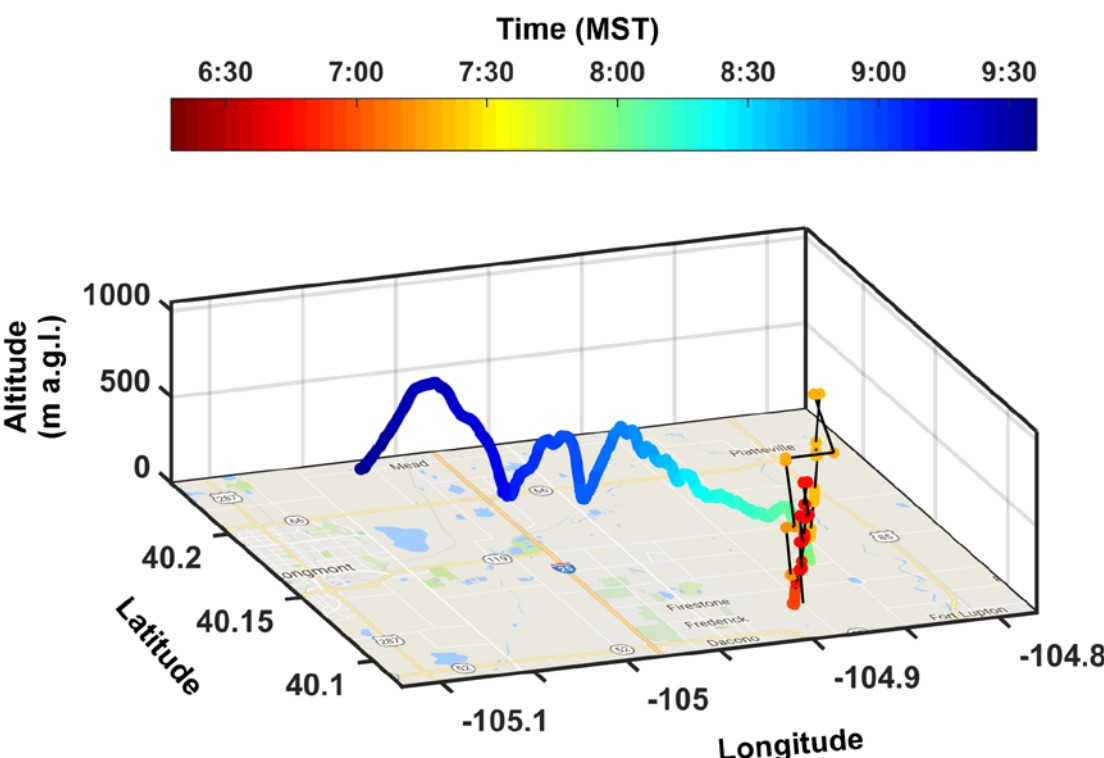

**Figure 2: Four-dimensional flight path of HOVERCAT during the 25 May 2017 test flight, coloured by time in Mountain Daylight Time (MDT). Black lines between data points indicate missing GPS data, which occurred between 7:01 – 7:07 and 7:23 – 7:51. Meters a.g.l. was calculated by subtracting 1490 from m a.m.s.l. to roughly show the altitude above ground.**

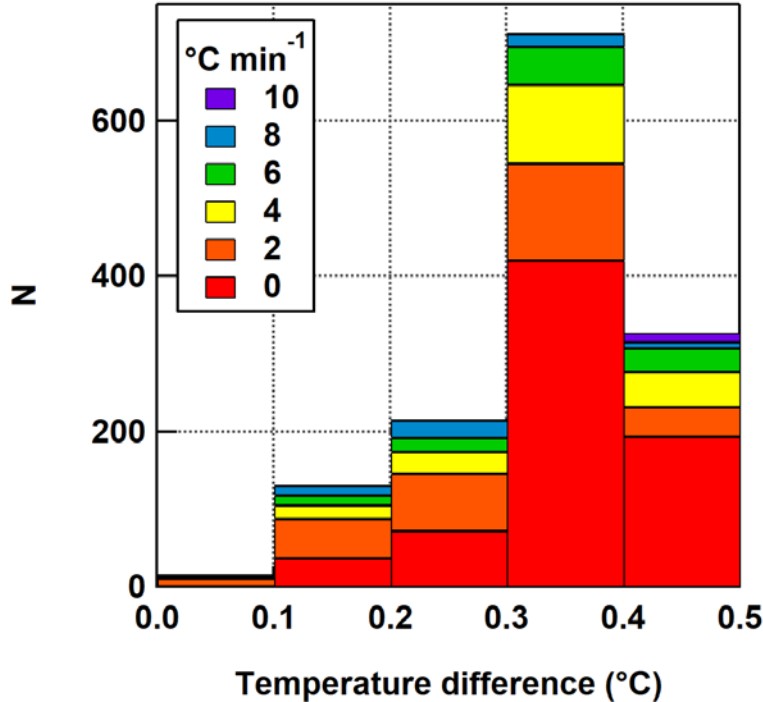

**Figure 3: Histogram of temperature differences between measurements from a probe at the centre of the copper plate and drop on top of the plate coated with petrolatum coloured by cooling rate. The 1-second data are from three different tests. The average difference used for the temperature correction was 0.33±0.15 °C.**

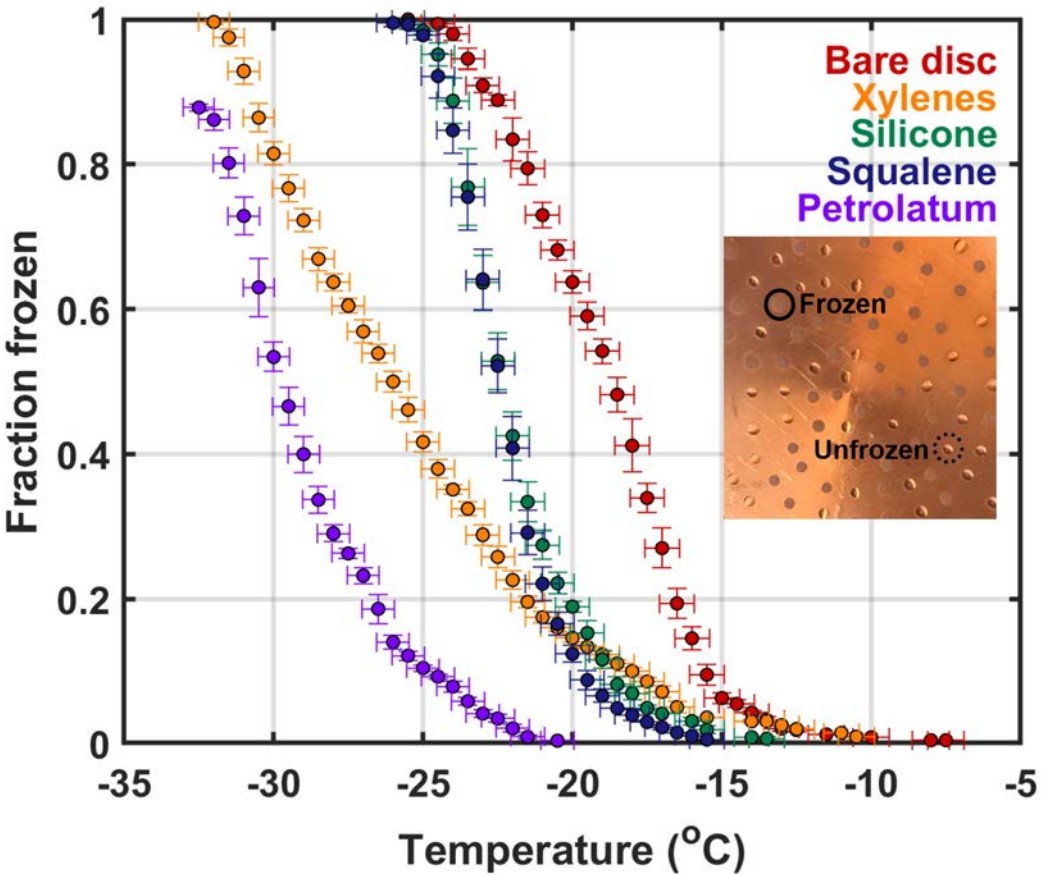

**Figure 4: Freezing spectra for the control experiments conducted to characterize the DFCP system. Results included here are tests evaluating the most proficient hydrophobic coating with blank UPW drops. Error bars for the y and x axes correspond to standard deviation per 0.5 °C bin and temperature probe/plate versus drop variability standard deviation, respectively. Spectra that do not reach a frozen fraction of 1 indicates not all drops froze at the lower limit of the DFCP. The inset shows an example of the appearance of frozen versus unfrozen 2.5-µL drops on the copper disc.**

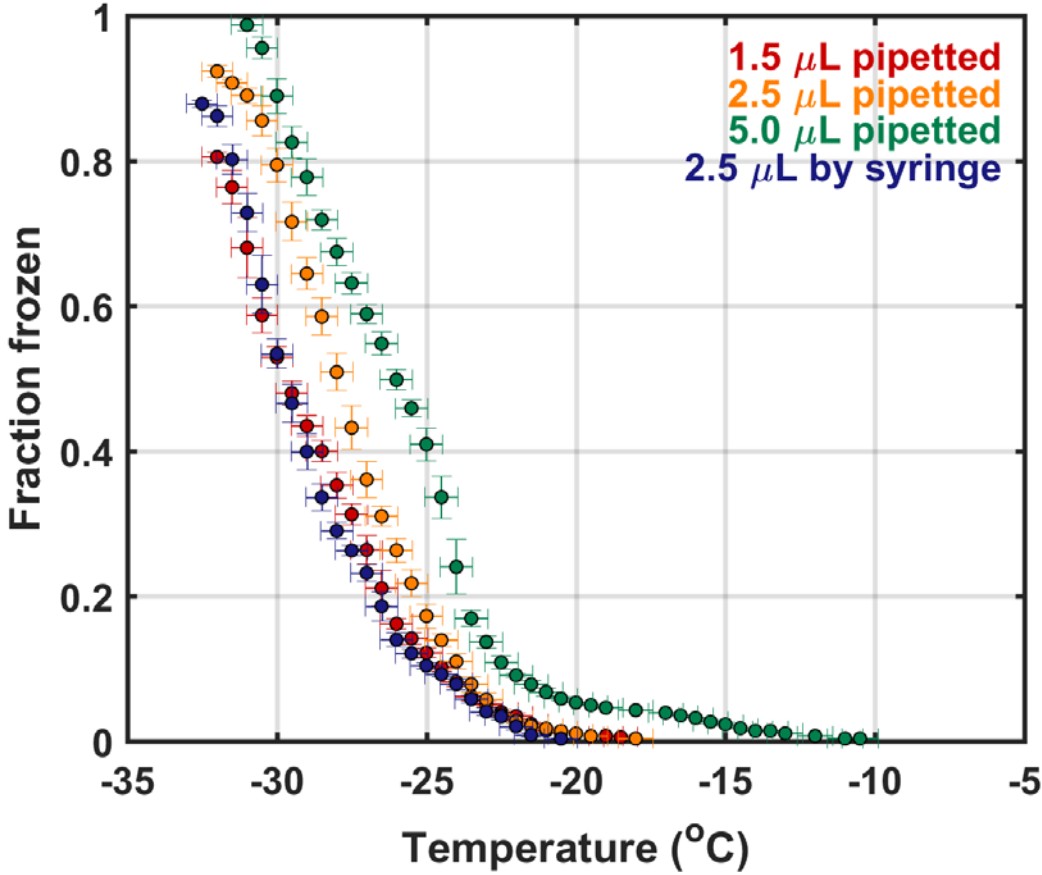

**Figure 5: Freezing spectra for the control experiments conducted to characterize the drop size chosen for DFCP analysis. Results included here are tests evaluating pipetted versus syringe-aliquoted drops and at different volumes. Error bars for the y and x axes correspond to standard deviation per 0.5 °C bin and temperature probe/plate versus drop variability standard deviation, respectively. Spectra that do not reach a frozen fraction of 1 indicates not all drops froze at the lower limit of the DFCP.**

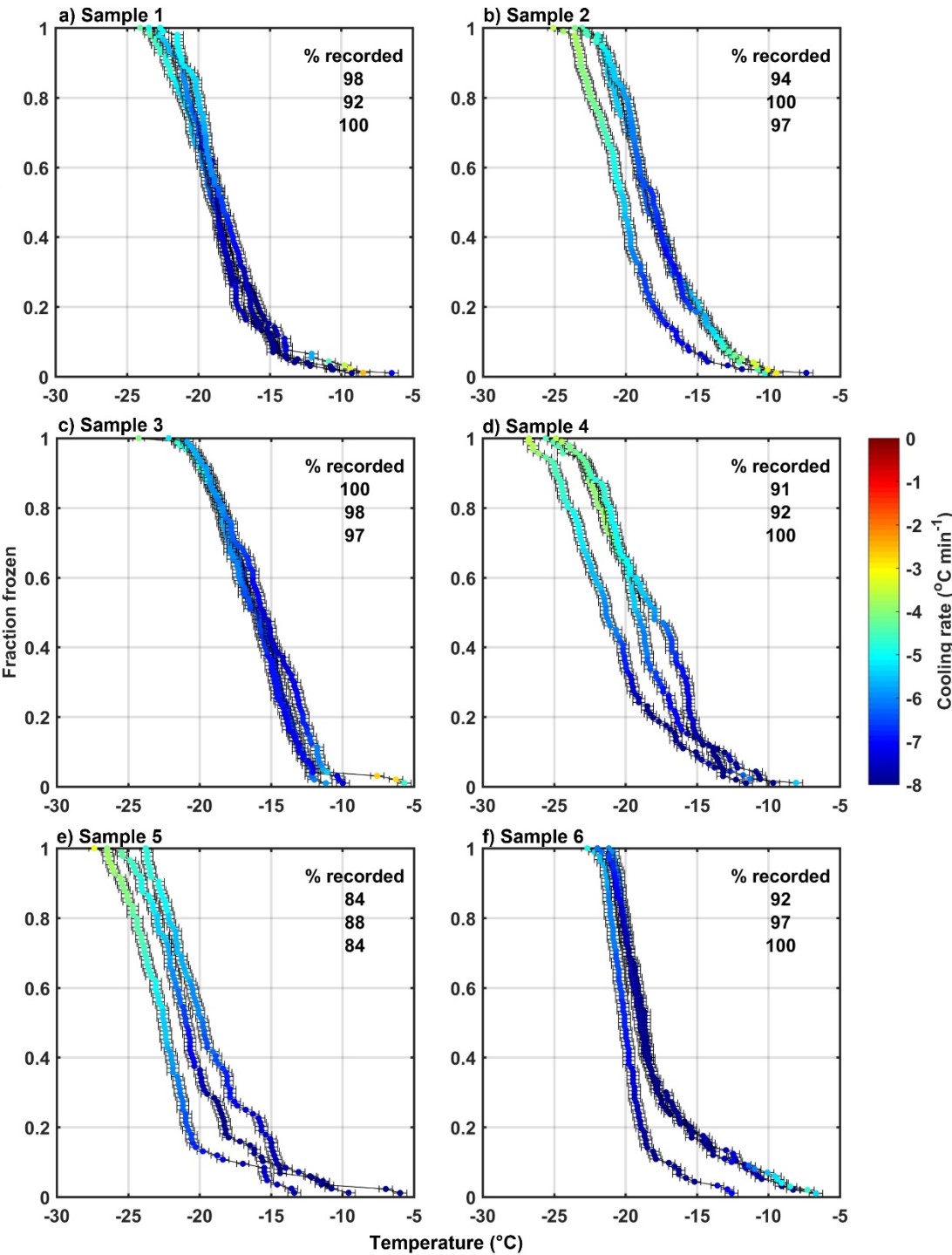

**Figure 6: Freezing spectra for the three tests of each of the samples collected from HOVERCAT during the 25 May 2017 test flight. Each data point is coloured by cooling rate and has error bars associated with Omega temperature probe uncertainty. The percentage of recorded frozen drops is provided for each sample.**

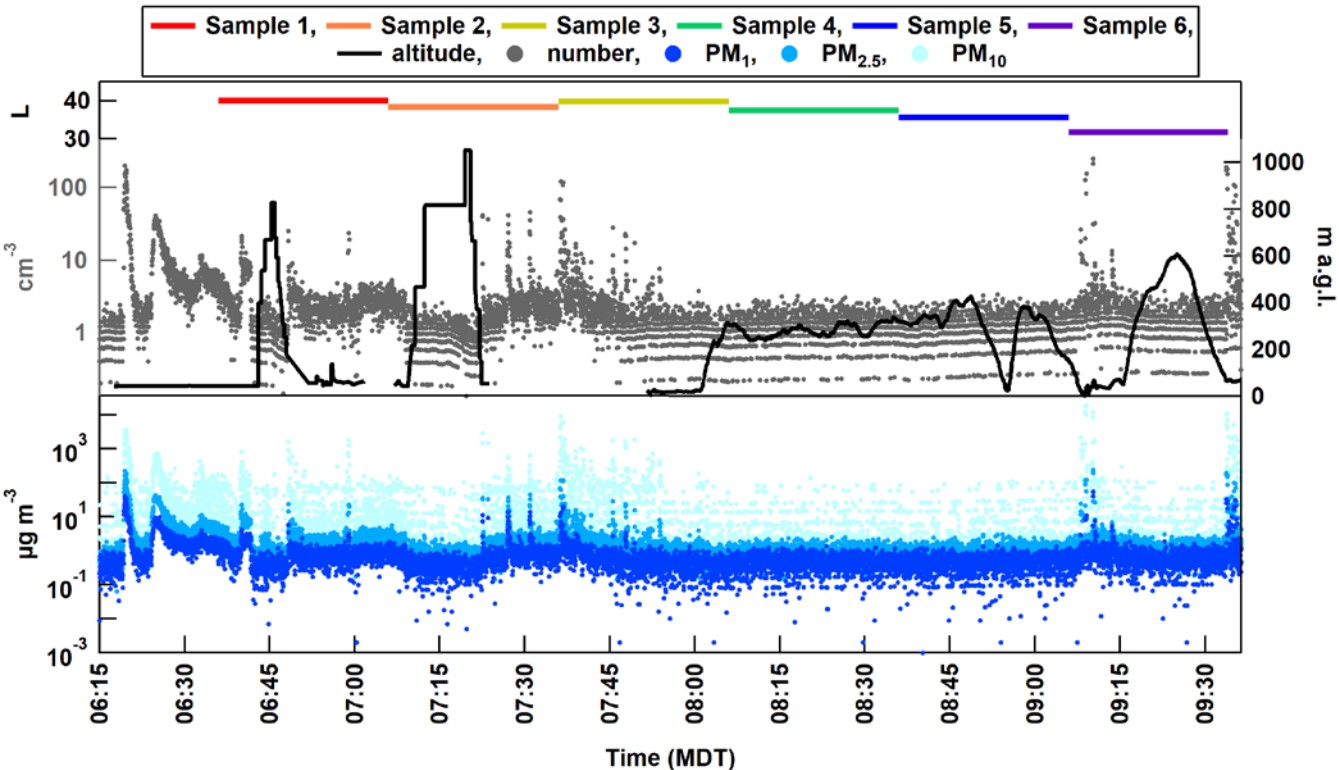

**Figure 7: Time series of TRAPS total volume per sample (L; of air), OPC number concentrations (cm⁻³), altitude (m a.g.l.), and estimated particulate mass (PM) concentrations from the OPC (µg m⁻³). The width of the TRAPS total volumes corresponds to the collection time per sample (i.e., 30 minutes).**

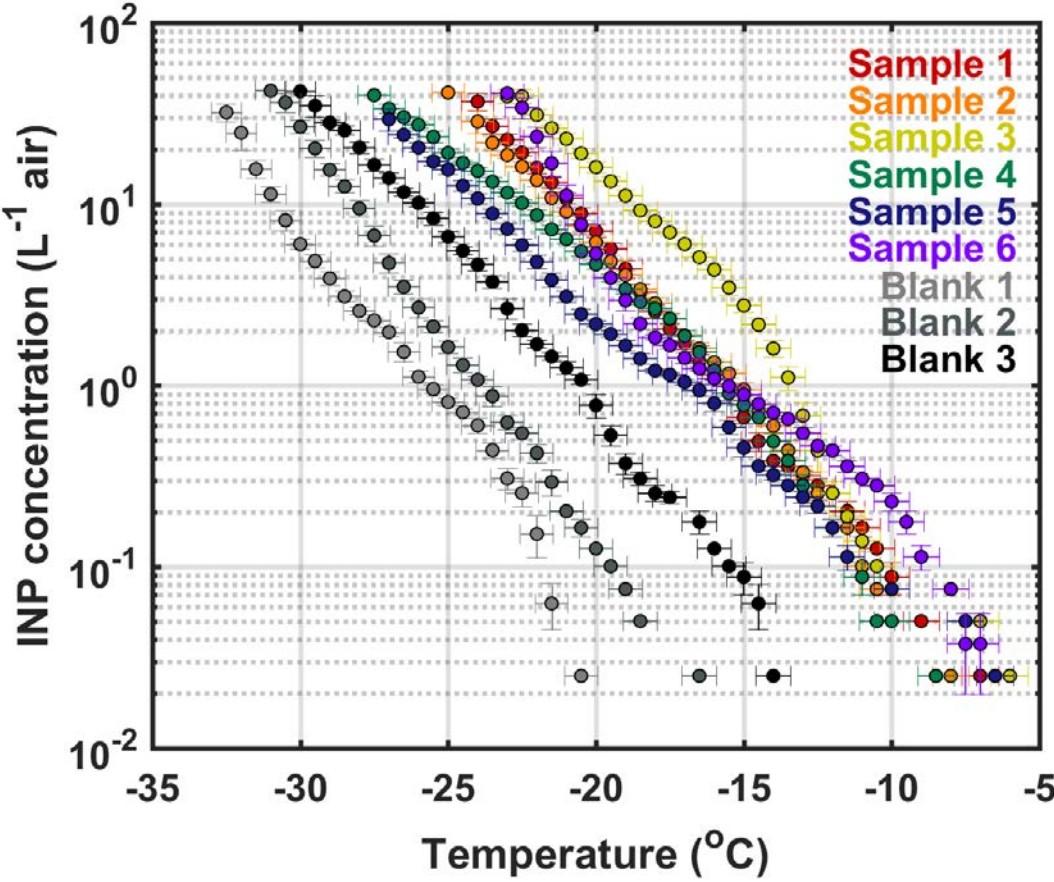

**Figure 8: Cumulative INP spectra from the samples collected during the 25 May 2017 HOVERCAT test flight. Triplicate tests are binned every 0.1 °C. The blanks indicate a triplicate test from: UPW mixed alone in a beaker for 2 hours (Blank 1), UPW mixed in a WhirlPak® bag for 2 hours (Blank 2), and an EmFab® filter mixed in UPW in a WhirlPak® bag for 2 hours (Blank 3). The latter is closest to how the samples were prepared. Error bars for the y and x axes correspond to standard deviation per 0.5 °C bin and temperature probe/plate versus drop variability standard deviation, respectively.**

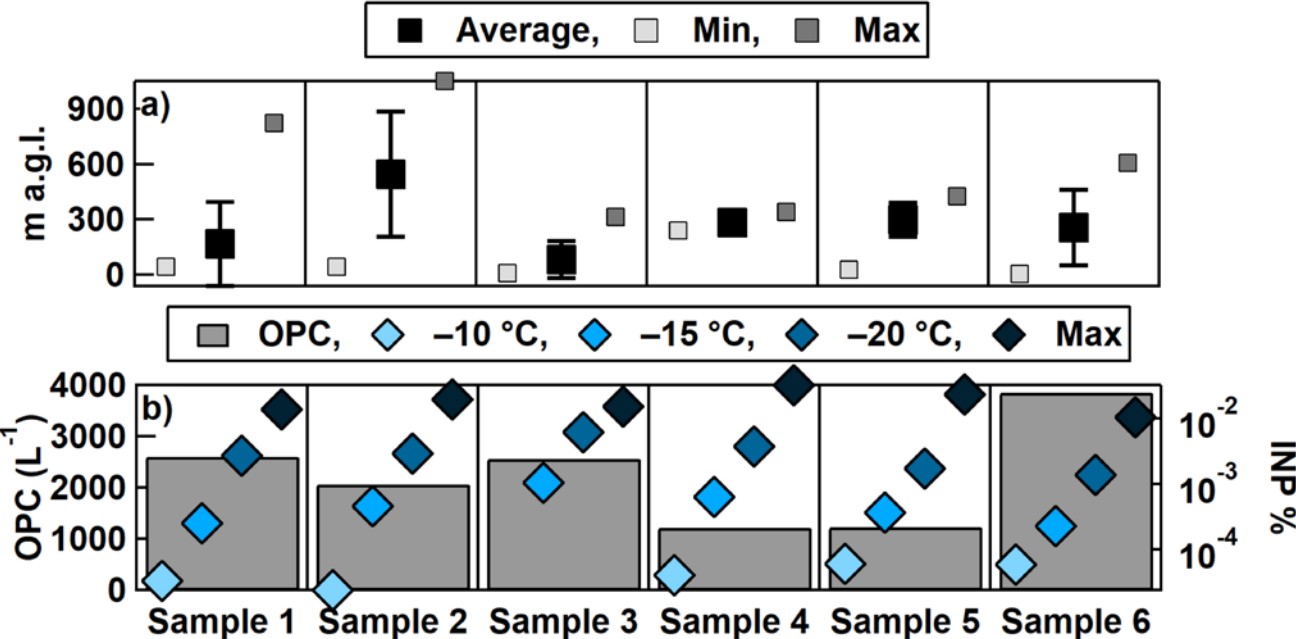

**Figure 9: a) Average, minimum, and maximum altitudes HOVERCAT flew during each sample collection time period. Error bars represent one standard deviation. b) The average number concentrations of total particles from 380 nm – 17 μm in diameter measured by the OPC (left axis) and fraction of INPs out of total OPC number at −10 °C, −15 °C, −20 °C, and the maximum INP concentration measured at the temperature in which the last drop froze (right axis).**

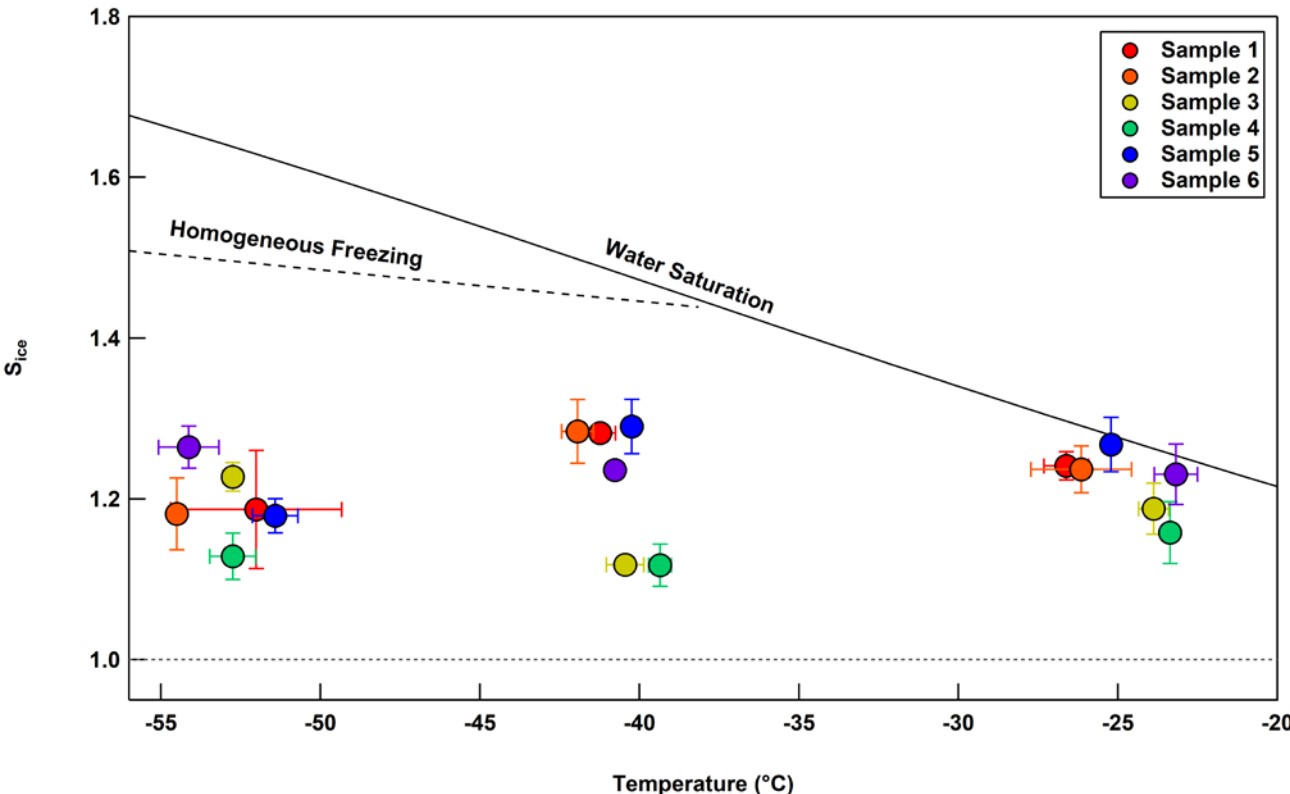

**Figure 10: Depositional ice nucleation experiments on Samples 1 – 6 plotted by $S_{ice}$ versus temperature. The values plotted here are of the onset conditions of depositional ice nucleation. For our experiments, this refers to the first particle to nucleate ice out of the $10^4$ particles deposited on the disc in total, thus a percent activated fraction of $10^{-4}$. Although temperatures measured were not exactly −25 °C, −40 °C, and −55 °C, these values are used for brevity for all samples within each grouping shown above. Nucleation occurring at −25 °C could also be due to immersion freezing.**

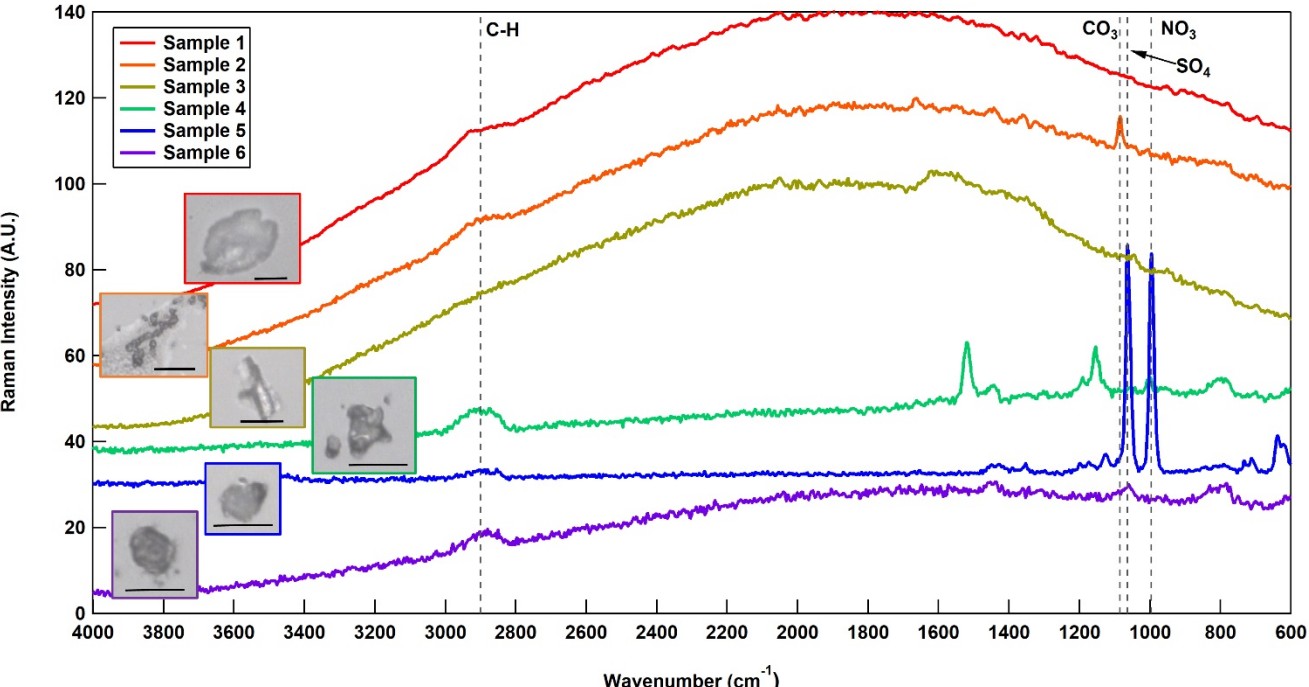

**Figure 11: Raman spectra for a representative particle per sample. Characteristic vibrational frequencies for functional groups of organics (C-H; 2800 – 3000 cm⁻¹), carbonates (CO₃; 1070 – 1090 cm⁻¹), sulphates (SO₄; 972 – 1008 cm⁻¹), and nitrates (NO₃; 1032 – 1069 cm⁻¹) are noted for reference. Included are images of the particles that initiated depositional freezing for the Raman spectra shown. The length of the black line in each image represents a scale of 20 μm.**

