# Peer review of "HOVERCAT: A novel aerial system for evaluation of aerosol-cloud interactions"

_Atmospheric Measurement Techniques, 2018_

## Referee Comment (RC1) · Anonymous Referee #1 · 16 Mar 2018

The manuscript by Creamean et al. (2018) introduces a novel measurements platform by which vertical profiles of aerosol size distributions (OPC) and multiple filter samples for ice nucleating particle (INP) analyses can be obtained. Furthermore, the authors characterize a newly build droplet freezing assay (DFPC – Droplet Freezing Cold Plate) and examine the uncertainty in temperature, test for dependencies in cooling rate and droplet size, and evaluate several hydrophobic coatings for the substrate. The second part of the INP analysis featured a deposition ice nucleation experiment coupled with Raman microscopy.

INP filter samples were collected with the newly build TRAPS (Time-Resolved Aerosol Particle Sampler), which was attached to HOVERCAT (Honing On VERtical Cloud and Aerosol properTies). HOVERCAT is a retrievable balloon-based measurement plat-

form, which theoretically should be able to hover at a constant altitude by the use of positive (sand ballaster) and negative (lift-gas) buoyancy adjustments. However, the manuscript presents only one successful test flight (out of only three existing flights). Even in this flight the altitude control has had its problems and could not perform the scheduled flight plan due to unfavorable weather conditions. As a result, during four out of six of the sampling periods the system was unable to maintain a constant altitude. This is the biggest problem with the manuscript. The publication of this manuscript seems a little hastened. First (?) test flights were conducted in late May 2017, then the collected samples were frozen for about six months, and the manuscript has been submitted in early February 2018. HOVERCAT (and the manuscript in turn) would have benefitted from some more extensive tests to really proof the principle (and name-giving signature ability) is reliably working. The authors know about this caveat as they emphasize that HOVERCAT in its current state is considered to be a prototype in the experimental development Phase I. Yet, I am not sure whether or not this classification as Phase I alone justifies the publication of the manuscript in this state.

The manuscript itself is well written and fits into the scope of the journal. The figures and scientific methods are of good quality. As the authors have correctly discerned there is a great need for more airborne INP measurements, since ice actually forms in clouds and not at the ground, where still most INP measurements are conducted. Here, the manuscript provides a valuable contribution to the community by introducing a promising alternative to "regular" aircraft-based missions. This leaves me with ambiguous feelings about a recommendation to publish the manuscript.

I encourage the authors to resubmit a revised manuscript that demonstrates the applicability of HOVERCAT under flight technical aspects more convincingly than it does at present. One more successful flight showing that the system is indeed able to sample the atmosphere at prescribed levels should be presented (maybe as a supplement without the INP analysis). In its present form the manuscript is a very interesting contribution to AMTD, yet of pre-print character.

General/Major Comments:

1) As mentioned above HOVERCAT's ability to hover at a constant altitude could not be reliably demonstrated (Figs. 2 and 7). Only 2 out of 6 samples show stable enough altitude conditions (one of which was collected close to the ground). Page 6, Line 3 states the original flight plan was a 5-step altitude profile with 500 m steps each. Yet, the actual flight profile looks nothing like that. Section 3.1 explains that the plan could not be fulfilled because of too windy conditions and a time delay between command and execution of buoyancy adjustments. The reader is then reassured that on a calmer day it might work. Yet, it leaves the reader wondering. How to achieve it in the future? Why didn't the authors do more flights until, the flight plan worked as planned? The rate of 1 out of 3 flights producing high enough sample volumes to be analyzed for INPs, seems also to be improvable. Have there been more test flights in the meantime?

In Section 3.4, where the future directions of HOVERCAT are listed, point 2) says: "operate successfully on a routine basis", however, I feel this should be a prerequisite for the platform, before thinking about publishing. It feels like step two is done before step one. In the Conclusion the platform is stated to have "the capability to hover at desired altitudes, making it an ideal system to collect sufficient aerosol loadings at a range of altitudes up to 2.6 km AMSL". The system may be capable of doing it (as sample 4 might suggest), yet I don't feel like it has proven without a doubt.

2) On a more technical note: Why didn't you sync the profiles with the sampling times? Wouldn't it be better to have a flexible (e.g. command based) way to communicate with the sampling system to adjust for difficulties and only start the pump when the profile is stable? Alternately, why not include a buffer of some minutes between prescribed sampling periods to account for maneuvering to reach the desired altitude?

3) In the Introduction the authors describe why more airborne INP measurements are needed and that there is a lack of alternatives to aircraft-based missions to gain vertically resolved information about INP. In this regard the paper by Schrod et al. (2017)
should definitely be cited and summarized. Schrod et al. (2017) describe the first INP measurements made onboard of an unmanned aircraft system (UAS) and follow a similar train of thought. On a similar note, the paper by Ardon-Dryer et al. (2011) should be mentioned as well. In it INP samples were collected at the ground and from a tethered balloon in Antarctica.

4) Fig. 6: Why didn't you wait until all drops are frozen – is it due to instrument limitations? Also, Page 7 Line 28f says the measurements were continued, until all droplets were frozen. Which fraction is used for calculating the INP concentration by the Vali equation (Fig. 8)? The ones shown in Fig. 6 or the actual number of frozen droplets relative to the total number of droplets? If the frozen fraction as shown in Fig. 6 were used, you'd create a bias towards a more ice active sample. I would rather show the plots as done in Figs. 4 and 5 (with the frozen fraction not normalized to 100%) to avoid confusion about this matter.

5) Fig. 10: Does it show the onset conditions of the first observed ice nucleation? Or is it a constant activated fraction of X %? Is there any quantitative measure of this ice nucleation technique available at all? From what I understand from section 2.3.2 it is not (at least in this study). Therefore, I would advise to be careful in making too strong statements, when just the first ice activation was observed. Go through the paragraph with that in mind. E.g. rewrite the sentence on Page 12 Line 1ff to something like this: "Overall, ice activation onset conditions between the six samples were similar at all temperatures tested. However, at -40°C Samples 3 and 4 showed first ice nucleation activity at saturation ice ratios X to X % earlier than the other samples and may be characterized as more efficient deposition INPs at that temperature as compared to the remaining samples. However, it should be emphasized that the presented results are not of a quantitative measure."

6) My impression is that sections 3.2 and 3.3 are somewhat overanalyzed in terms of trying to explain the sources etc. After all, the result section of this manuscript depicts only six consecutive 30 minute samples of one day, most of which feature

a considerable variance in sampling altitude, which should make it difficult to make general statements about the different samples. Also only 3 – 5 INPs were analyzed with Raman spectroscopy from each sampling spot, again making it difficult to allow general statements about each sample. I will give some examples: a) Page 11, Line 10ff: I think this statement is somewhat misleading. Sample 6 had only a short period of approx. 5 minutes where it was hovering close to the ground. The corresponding reasoning is rather speculative. b) Page 11, Line 17f: 1) and 2) are identical, but only phrased differently? c) Page 11, Line 18f: Although it is a logical statement, is this really supported by the data at this point? d) Page 12, Line 5f: It should be avoided to speak too generally here, since only 3-5 particles per sample were analyzed. How this few particles be representative of a whole sample? e) Page 12, Line 15f & Page 12, Line 17ff: This is again rather speculative.

Specific/Minor Comments:

1. The terming of HOVERCATS ability to measure "time-resolved" INP concentrations (Abstract, Page 13 Line 11) seems a little misleading, since it averages over broader time periods instead of measuring in real-time.

2. Use meters above ground level throughout the manuscript (other than when discussion problems related to the flow of the pump). The Colorado plains has a rather high ground elevation, which may lead the reader to think the balloon was flying higher than it actually was.

3. Page 2, Line 10: When listing the ice nucleation modes, a reference to Vali et al. (2015) should be considered. Also, the concept of pore condensation freezing as introduced by Marcolli et al. (2014) could be mentioned, when describing deposition freezing.

4. Describe in more detail how the system adjusts the altitude, how it lands and how it is retrieved. Is the balloon still intact after landing, can it be reused?

[Figure]

5. Why haven't you used a more powerful pump that can go above 1.2 lpm? I understand that payload restrictions are crucial for this approach, but I think the system would be much more flexible when the sample flow could go higher.

6. Page 10, Line 28ff: I think it would be better to give median (or average) particle concentrations instead of the maximum, since near ground values showed episodic peaks. Also, a plot showing the particle concentration vs. altitude could be worthwhile.

7. I wonder if you considered to calculate the correlation between OPC concentration larger 0.5 $\mu$m (as in DeMott et al., 2010, 2015) and the INP concentration at T = –X °C averaged over each sampling period (although N = 6 isn't very good statistics), and/or add a scatter plot.

8. Page 11, Line 12: Give the range of minutes (from X to X minutes) that HOVER-CAT hovered near the ground. Following the reasoning on Page 11, Line 14ff it may be worthwhile to correlate the number of minutes close to the ground with the INP concentration of the corresponding sample.

9. In section 3.4 the authors should add a goal to improve on the ability to stay at a constant altitude with HOVERCAT.

10. Add a Figure that shows a detailed schematic of the TRAPS. I see that a similar Figure can be found in Ogren et al. (2017). Yet, I feel it is essential to this manuscript as well and should be added therefore (also the design probably is different to what is shown in Ogren et al. (2017)). A few sentences describing the filter collection with TRAPS in more detail could also added to the text.

11. Fig. 2: Indicate the exact sampling times (e.g. using a black sphere to mark the beginning of a new sample on the flight track, and a black line one the time color scale)

12. Fig. 3: Y-axis and color scale give the same information, one of those seems redundant. A stacked (color-coded by cooling rate) histogram might give more information (or as additional Fig. b). Something like my Figure 1 attached to this review

(not your data):

13. Fig. 7: Give the altitude in m AGL (starting from 0). Use more distinct colors for altitude and OPC (instead of light and dark grey). Coloring for Sample 2 is not the same orange as in the others figures. Maybe use logarithmic scale for the OPC concentration?

14. Fig. 10: Add information to the labeling describing what exactly is depicted on this figure. Is it the onset conditions of the first observed ice nucleation event (or of X % activated fraction)?

15. Fig. 11: What is meant by "most representative particle type per sample"? In section 2.3.2 it says that only 3 – 5 particles were analyzed with Raman spectrometry per sample. How do you know which particle type is representative for the whole sample when only 3 – 5 particles each were analyzed in total?

Technical Comments:

Page 2, Line 17: Replace "adroit" with "efficient"

Page 5, Line 10: Remove "-" in "1.2-L min-1"

Page 5, Line 13: Add space between "12 V DC"

Page 10, Line 23ff: The term "profile" is a little confusing, since it does not correspond to the same samplings (e.g. profile 3 is sample 4). Maybe add a short sentence that the two terms are not the same.

Page 12, Line 16: Typo in reference "Möhler et al., 2008"

Fig. 1 Caption: b) Picture of the aerosol module.

Fig. 8: Labeling of "UPW in breaker" is covering part of the data. The last sentence of the caption does not make sense for this plot.

References:

Ardon-Dryer et al. (2011). Ardon-Dryer, K., Levin, Z., and Lawson, R. P.: Characteristics of immersion freezing nuclei at the South Pole station in Antarctica, Atmos. Chem. Phys., 11, 4015-4024, https://doi.org/10.5194/acp-11-4015-2011, 2011.

DeMott et al. (2010). DeMott, P. J., Prenni, A. J., Liu, X., Kreidenweis, S. M., Petters, M. D., Twohy, C. H.,Richardson, M. S., Eidhammer, T., and Rogers, D. C.: Predicting global atmospheric ice nuclei distributions and their impacts on climate, Proceedings of the National Academy of Sciences of the United States of America, 107, 11217-11222, 10.1073/pnas.0910818107, 2010.

DeMott et al. (2015). DeMott, P. J., Prenni, A. J., McMeeking, G. R., Sullivan, R. C., Petters, M. D., Tobo, Y., Niemand, M., Möhler, O., Snider, J. R., Wang, Z., and Kreidenweis, S. M.: Integrating laboratory and field data to quantify the immersion freezing ice nucleation activity of mineral dust particles, Atmos. Chem. Phys., 15, 393–409, doi:10.5194/acp-15-393-2015, 2015.

Marcolli et al. (2014). Marcolli, C.: Deposition nucleation viewed as homogeneous or immersion freezing in pores and cavities, Atmos. Chem. Phys., 14, 2071-2104, https://doi.org/10.5194/acp-14-2071-2014, 2014.

Schrod et al. (2017). Schrod, J., Weber, D., Drücke, J., Keleshis, C., Pikridas, M., Ebert, M., Cvetković, B., Nickovic, S., Marinou, E., Baars, H., Ansmann, A., Vrekoussis, M., Mihalopoulos, N., Sciare, J., Curtius, J., and Bingemer, H. G.: Ice nucleating particles over the Eastern Mediterranean measured by unmanned aircraft systems, Atmos. Chem. Phys., 17, 4817-4835, doi:10.5194/acp-17-4817-2017, 2017.

Vali et al. (2015). Vali, G., DeMott, P. J., Möhler, O., and Whale, T. F.: Technical Note: A proposal for ice nucleation terminology, Atmos. Chem. Phys., 15, 10263–10270, doi:10.5194/acp-15-10263-2015, 2015.

[Figure]

[Figure]

**Fig. 1.**

---

## Referee Comment (RC2) · Anonymous Referee #2 · 8 Apr 2018

Review of HOVERCAT (AMT-2018-47): A novel aerial system for evaluation of aerosol-cloud-interactions by Creamean et al.

The authors present HOVERCAT a platform for airborne sampling of aerosol and particulate matter which can later be used for offline INP analysis. In particular they propose vertical profiling is possible by doing stepwise ascents in their balloon platform. This is indeed very exciting, very promising and the right direction for the field instruments to progress in.

I have some issues with the scarcity of test flights conducted and the rush to publish something. I fully support a publication addressing HOVERCAT, but I can't help but wonder if the manuscript would be more complete and exciting if further development that is already being done would be included herein? I would recommend publication after major revisions and the concerns below have been addressed. I do not separate major from minor concerns and technical comments from general scientific comments; instead, I list all concerns by page.

In general I think many discussion points are missing and some of the data is not sfucciently addressed in the text. Also some of the instrumental development aspects need to be addressed. Overall I think this paper will make a very good contribution to the literature and I think it should be published, but more needs to be done.

**Page 1**

Line 11. Precipitation processes "can" be modulated by aerosol cloud, not "are" since other dynamics also influence radiative and precipitation processes.

Line 15-20: Upon reading the manuscript and seeing how only 1 successful test has been presented herein, I would soften the ideas about "hovering" at as given altitude. It was clear that the authors struggled to hover at a given altitude because of vertical winds and the balloon could not respond fast enough to maintain altitude. I think one test flight is too soon to claim this.

For realtime aerosol measurements, the vertical profile would still be achieveavle, but for INP analysis as the authors rightfully acknowledge more time is needed at each sampling altitude to get enough PM on their filter, and this is harder to achievbe without fluctuations (sudden rises, or falls). Perhaps it would help if the authors stated a range xx +/- yy m hovering capability. So that instead of being sample to sample at say 500 M agl.. HOVERCAT can sample at 500 +/-100 m as an example.

**Page 2**

line 10. An appropriate reference to add or even to replace Cziczo et al. (2017)(who talk about IN methods as a focus not the mechanisms) with would Vali et al. (2015) who have a nice discussion that included the IN community on the terminology and IN mechanisms.

Line 12: immersion freezing is the most relevant heterogeneous ice nucleation mechanism for MPCs but not for cloud ice formation. We don't know that, there is a fair bit of literature to suggest that majority of cloud ice may come from secondary ice multiplication. So I suggest re-phrase to make this more clear.

**Page 3**

Line 1: here the authors say 2 km AGL is a limitation but then the platform they present is only shown to go to 1.1 km AGL, this is contradictory. I suggest correct this or modify the way this is presented.

Line 4 -5: there may have not been INP measurements on a balloon system, but a similar idea of collecting aerosol on filters/wafers on a UAV was already done couple years ago and published (Schrod et al., 2017). This study should be discussed and acknowledged here.

**Page 5**

Line 8: Maybe state what density/shape factor relationships were used to convert OPC data into PM values

Line 10: typo should be 1.2 L not 1.2-L

**Page 6**

This is a nice description of the flight and the testing done. Indeed, it is exciting to read that such developments are taking place. My only concern about this is if two out of the three test flights were not successful and if the altitude of 2.5 km AMSL was determined to be the max altitude range of operation, why weren't more flights conducted for this study, once the operational parameters were clear? This should have been straightforward to do, no? Furthermore, if stronger pumps are being tested for higher altitude (lower pressure) sampling why not wait for those tests so that a test flight with higher altitude can also be conducted.

Line 22-28: The data given in this paragraph is all for ground level or at a given height? I guess if the goal is vertical profiling, shouldn't the meteorological data for the different height be useful or may be needed to understand the aerosol properties (e.g. phase state) during sampling. Does it help to have all the ground based data when all the sampling is occurring above ground?

**Page 7**

Line 1: make clear 47 mm is diameter.

Line 25: here the authors say control experiments were done with ultrapure water at various cooling rates. How was the water treated here. In order to have a true control, the water should also have been subjected to the same process as the samples i.e.

" (i.e., samples) were successfully collected before the battery died. Each spot was placed is a 29-mL sterile Whirlpak® bag with 2 mL of UPW to resuspend particles deposited on the filter"

The water should be treated as above to ensure there is no INP contamination coming from the whirlpak bag. Have such experiments been done? If not, I suggest these should be included in order to really be confident that the freezing is from the aerosol re-suspended into the sample.

**Page 8**

Line 4: I understand how the temperature calibration experiment was done, but I tried hard but do not follow how the data are plotted in Figure 3. Which data are for the probe at the center of the copper plate and which for the drop on top of the stage? The vertical lines just mean that the temperature difference doesn't change with cooling rate, but then why are there so many vertical lines, and what determined the position of the data on the x-axis, this figure is difficult to follow. I also don't understand why there is a cooling rate on the y-axis and then the data are also colored by cooling rate?

Line 27, why does it take more time to prepare drops with a pipette. Are you using a single tipped pipette, but you could use an 8-channel or 12 channel pipette for this too. It would speed things up.

**Page 9**

Line 4. It was not clear to me why all 100 drops could not be recorded? If you place 100 drops on a cold stage what is the hindrance in recording data until all 100 drops are frozen?

**Page 10**

Line 10: based on your filter sampling, what was the efficiency of smapling particles larger than 10um? From the deposition mode experiments the authors say they see particles between 1 – 50 um. This could be a strong indication of coagulation of the aerosol based on the suspension and nebulization methods used. Does it then make sense to assume that one particle lead to each ice crystal observed? And also try and analyses that particle? It is likely a particle that is composed of multiple particles. With more samples perhaps the authors could do a particle count by feeding the nebulized spray though an SMPS or even into a CPC to get a particle count. And then compare this to a particle count taken from image analysis of the deposition state to see if numbers are similar. This will provide some correction factor (if needed) for how much coagulation maybe taking place when the particles are being deposited onto the stage for deposition nucleation.

Line 25-30: regarding sudden drops and hovering – see comment above that I made in abstract section. I think it is too soon to claim the ability to hover at a fixed altitude. Perhaps consider putting a range to that, given that the balloon cannot respond fast enough to the updrafts experienced. I still think this method is super valuable even for example you say, each INP filter sample is conducted within a range of 100 m or so. This is a step in the right direction for vertical profiling.

**Page 11.**

Section 3.2. Results are indeed interesting, but it would have been nice to see that each sample presented did not have overlapping altitudes. Is there no way to automatically shut off filter sampling when a certain threshold of a sudden rise or fall is crossed and then only resume sampling when the balloon is within this threshold again? Would this be too demanding, or is there not enough time in the flight for this. This way you ensure that your INP filter samples are restricted to sampling at a certain altitude and you can resolve better the vertical profiling of the INPS. For example in Figure 9, sample 2, if the sampling was shut off for the periods when the instrument was below 600 m AGL, that would allow sample 2 to be representative of only INPs above 600m AGL.

Section 3.3:  should be deposition nucleation – not freezing, freezing implies liquid to solid transition, I think the authors' intent is to imply vapour to ice mechanism here, therefore I would take freezing out of the phrase.

**Page 12**

Line 1: "…above homogenous freezing" are you referring to temperature or Si? Could clarify here already. Based on Fig. 10 it looks like you refer to both. However, some of your data points are quite close to water saturation (-25 for example), so it maybe a tough sell to claim those as be depositional ice nucleation.

Section 3.3: I like the discussion in this section. But here one must acknowledge that there could be artifacts of having your INPs as agglomerates of particles. So maybe the aerosol was externally mixed, but with the experimental method used, they could become or appear as internally mixed since you suspend and nebulize and then allow for evaporation to retain the residual particles on the cold stage. This should allow for some coagulation, and perhaps the Raman is investigating a single particle that is a result of multiple smaller coagulated particles.

Section 3.4: Again while reading this section, I can't help but feel that the paper is a little premature. Some of the plans and developments described herein could be already part of this manuscript, like the stronger pumps, a few more test flights, certainly more than 1, and the practice to control the balloon to stay at desired altitudes. i.e., the above should be part of Phase I.

I understand regulatory approval work like compliance with FAA can be phase II as well as launching on other airborne platforms (UAS, reverse parachute). i.e. demonstrating their instruments are versatile enough for other platforms, can all be phase II.

**Page 13**

Line 5. Certainly if the plans to operate at Jungfraujoch in Spring 2018 are on track (that is now), then stronger pumps have been implemented already. Since ground level at Jungfraujoch is already ~ 650 mb pressure.

Line 14: maybe put in also the altitude AGL to give context to your starting point.

Line 19-20: Soften the vertical profiling statement here because the authors showed overlapping vertical profiles, so it is not yet achievable in the strictest of sense.

Given that this is an instrument development paper, I would like to see a section on Benefits and limitations of the instrument. Where the authors state this clearly. I believe some of this is interspersed through the manuscript, but I think this should be brought together in one section to make it clear what are the benefits of HOVERCAT (and its current limitations).

Figure 4. For clarity that the control experiments are done with pure water and also it would be good to indicate the volume of the drops in the caption.

Figure 5. I was a little confused – the "by hand" drops, are those not the same as the syringe drops? Or how were the 2.5 ul drops by hand produced? There must have been some sort of tool for these.

Figure 7: Could you make it clear in the caption that the grey line plot is the one corresponding to the altitude and the scatter plot corresponds to the concentration from the OPC? Is it mentioned what densities or shape factors have been assumed to come up with the PM values from the OPC data?

Were the particles losses of 10% accounted for in the results? i.e. when calculating the concentrations in Fig. 8. Also, how were the control experiments for the ultra pure water in beaker and bag converted to INP concentrations? These fraction curves are not addressed sufficiently in the text I think.

References

Cziczo, D. J., Ladino, L., Boose, Y., Kanji, Z. A., Kupiszewski, P., Lance, S., Mertes, S., and Wex, H.: Measurements of Ice Nucleating Particles and Ice Residuals, 58, 8.1-8.13, doi:10.1175/AMSMONOGRAPHS-D-16-0008.1, 2017.

Schrod, J., Weber, D., Drücke, J., Keleshis, C., Pikridas, M., Ebert, M., CvetkoviÄ‡, B., Nickovic, S., Marinou, E., Baars, H., Ansmann, A., Vrekoussis, M., Mihalopoulos, N., Sciare, J., Curtius, J., and Bingemer, H. G.: Ice nucleating particles over the Eastern Mediterranean measured by unmanned aircraft systems, 17, 4817-4835, doi:10.5194/acp-17-4817-2017, 2017.

Vali, G., DeMott, P. J., Möhler, O., and Whale, T. F.: Technical Note: A proposal for ice nucleation terminology, 15, 10263-10270, doi:10.5194/acp-15-10263-2015, 2015.

---

## Author Comment (AC1) · 23 May 2018

We would like to thank the reviewers for their insightful and helpful comments, and encouragement for the need of vertical profiling of INPs. As a result of revision based on their feedback, the manuscript is much stronger. Below are the responses to reviewer comments in blue. We note that both reviewers were generally concerned with the hovering capability of the system in addition to presenting one successful flight out of three. As a result, we have significantly revised the tone and organization of the manuscript, in addition to providing the community with what did and did not work, and recommendations for those interested in developing similar systems in the future. More details and specific examples of where text was modified is provided in the responses below. We have attached a track changes version for reference of the specific revisions made.

**Reviewer 1**

**General/Major Comments:**

1) As mentioned above HOVERCAT's ability to hover at a constant altitude could not be reliably demonstrated (Figs. 2 and 7). Only 2 out of 6 samples show stable enough altitude conditions (one of which was collected close to the ground). Page 6, Line 3 states the original flight plan was a 5-step altitude profile with 500 m steps each. Yet, the actual flight profile looks nothing like that. Section 3.1 explains that the plan could not be fulfilled because of too windy conditions and a time delay between command and execution of buoyancy adjustments. The reader is then reassured that on a calmer day it might work. Yet, it leaves the reader wondering. How to achieve it in the future? Why didn't the authors do more flights until, the flight plan worked as planned? The rate of 1 out of 3 flights producing high enough sample volumes to be analyzed for INPs, seems also to be improvable. Have there been more test flights in the meantime?

We have softened the hovering aspect of the system throughout the manuscript to rectify the issue with the hovering flight plan. We also added, "However, this plan was ultimately not executed due to flight complications discussed herein." to section 2.4. The only places the word "hover" remains in the manuscript is in the HOVERCAT acronym. We also revised the definition of the BBFCS system such that the user can control the ascent and descent rates of the balloon instead of "controlling the altitude".

We realize we were not initially clear with what components were included in the HOVERCAT definition, which we intended to define as the aerosol components, since the honing on vertical cloud and aerosol properties is the responsibility of the aerosol instrumentation. We have now made this clear throughout the manuscript.

To address the issue with having only one successful flight, we note that the system *does* work, but with its caveats. Even the Ardon-Dryer et al. (2011) study that the reviewer mentions report only three filters from one day, indicating a substantial number of flights and samples is not required for publication of what (very) limited INP profiles exist. We emphasize that this is Phase I and think it is important to discuss what *didn't* work as that is important to other researchers who may be interested in developing parallel systems in the future, such that they will not have to "reinvent the wheel" so to speak. This is the reasoning behind keeping the original flight plan, such that our idea of stepping is not as easy as one might think. However, we now modified section 3.4 to discuss our recommendations for those interested in controlled launched balloon measurements, including a list of what to do and not to do. This addition provides rationale for keeping in the flight plans, why we had one out of three successful flights, and provides suggestions for improvement. There is a certain stigma with publishing only partially successful results or deployments, but we aim to break that mold and show what didn't work to help those in the future. In addition, we originally intended to showcase the vertical profiling of both immersion and deposition mode INPs with the aerosol module, which is indeed novel, but have now emphasized that throughout the manuscript.

To provide some specific examples of where we modified the text:
- We revised the abstract to highlight the aerosol module and state that we provide recommendations for use of future launched platforms.
- We instead emphasize that the BBFCS has a benefit of being more cost effective and flexible (in terms of FAA regulations, launch locations, and personnel required) than traditional tethered platforms such that it does not require a winch and that we can slow the ascent rate in addition to providing a descent option which, combined, is not a feather attainable with traditional launched balloons. This information is provided in section 2.1.

- We completely revised and reorganized the methods section to clearly define HOVERCAT and describe that first, followed by describing but not showcasing the balloon platform.
- We revised section 3.4 to emphasize our scientific priorities (discussed in detail in our response to comment 2) and recommendations for future measurements.

In Section 3.4, where the future directions of HOVERCAT are listed, point 2) says: "operate successfully on a routine basis", however, I feel this should be a prerequisite for the platform, before thinking about publishing. It feels like step two is done before step one. In the Conclusion the platform is stated to have "the capability to hover at desired altitudes, making it an ideal system to collect sufficient aerosol loadings at a range of altitudes up to 2.6 km AMSL". The system may be capable of doing it (as sample 4 might suggest), yet I don't feel like it has proven without a doubt.

This section was considerably revamped to instead discuss recommendations for future deployments of such systems in general. We agree that it has not been proven without a doubt that the system can hover—this concern was alleviated by toning down the hovering capability description throughout the entire manuscript.

2) On a more technical note: Why didn't you sync the profiles with the sampling times? Wouldn't it be better to have a flexible (e.g. command based) way to communicate with the sampling system to adjust for difficulties and only start the pump when the profile is stable? Alternately, why not include a buffer of some minutes between prescribed sampling periods to account for maneuvering to reach the desired altitude?

Certainly, it would be easier to enable ground-based bidirectional commands to change the sample spot when desired. However, the manuscript presents a pilot study focusing on specific priorities: (1) Can we recover the system? And (2) Can we control the ascent and descent? Both priorities fall under the umbrella of making a system that is relatively affordable, user friendly, and ensuring avoidance of complicated FAA COA paperwork. Adding complexity to the system for controlling altitude and sampling frequency adds cost, weight, and more variables that could fail. For clarity, we now emphasize that this was a pilot study throughout the manuscript (in addition what we already call "test" flights) and discuss our priorities in section 3.4.

More specifically, a bidirectional communication feature entails a significantly more advanced ground-based system and would add too much weight to HOVERCAT through additional processing hardware, an additional receiver, and batteries needed to control in such a manner. First, the BBFCS had bidirectional communication, but additional channels would need to be added to control the TRAPS as well, adding to the cost. And second, adding the necessary components to the TRAPS would add to both cost and weight, and given our 400 g buffer (i.e., calculated from the 2700 g limit and our HOVERCAT 2300 g weight), would likely push the system into a weight category that requires either restricted airspace or significant paperwork to obtain and FAA COA to fly outside of restricted airspace areas. Both of these issues make flying HOVERCAT more complicated and expensive, which fall outside of our vision to make an affordable and easy-to-use system.

3) In the Introduction the authors describe why more airborne INP measurements are needed and that there is a lack of alternatives to aircraft-based missions to gain vertically resolved information about INP. In this regard the paper by Schrod et al. (2017) should definitely be cited and summarized. Schrod et al. (2017) describe the first INP measurements made onboard of an unmanned aircraft system (UAS) and follow a similar train of thought. On a similar note, the paper by Ardon-Dryer et al. (2011) should be mentioned as well. In it INP samples were collected at the ground and from a tethered balloon in Antarctica.

Thank you for bringing these references to our attention, which are certainly important and noteworthy studies with regarding to the vertical distribution of atmospheric INPs. We have added both Schrod et al. (2017) and Ardon-Dryer et al. (2011) to the introduction, their advantages, and how our results differ from theirs. Schrod et al. (2017) measured INPs but only in the deposition mode. Ardon-Dryer et al. (2011) present results more relevant to ours, but only measured INPs below −18 °C in immersion mode on 3 samples and only up to 196 m a.g.l. The fact that only two studies before ours exist, and the limitations of those studies and ours, demonstrates the challenges associated with obtaining vertically profiled INPs, which is important to highlight and why we directly state this after describing their work in the introduction. We also removed the statement, "Additionally, to our knowledge, there have been no measurements of INPs via any balloon platform."

4) Fig. 6: Why didn't you wait until all drops are frozen – is it due to instrument limitations? Also, Page 7 Line 28f says the measurements were continued, until all droplets were frozen. Which fraction is used for calculating the INP concentration by the Vali equation (Fig. 8)? The ones shown in Fig. 6 or the actual number of frozen droplets relative to the total number of droplets? If the frozen fraction as shown in Fig. 6 were used, you'd create a bias towards a more ice active sample. I would rather show the plots as done in Figs. 4 and 5 (with the frozen fraction not normalized to 100%) to avoid confusion about this matter.

We realize our description was confusing. At times, not all drops froze for the control experiments with UPW due to the system detection limit (i.e., it can only go down to approximately −32 to −33 °C). The fraction frozen was the number of frozen droplets divided by the number of droplets detected + unfrozen drops, which is why we did not always observe a frozen fraction of 1. However, all drops froze from the blanks with the bag, blanks with the filter, and the collected samples themselves. We more clearly define this in section 2.3.1.

5) Fig. 10: Does it show the onset conditions of the first observed ice nucleation? Or is it a constant activated fraction of X %? Is there any quantitative measure of this ice nucleation technique available at all? From what I understand from section 2.3.2 it is not (at least in this study). Therefore, I would advise to be careful in making too strong statements, when just the first ice activation was observed. Go through the paragraph with that in mind. E.g. rewrite the sentence on Page 12 Line 1ff to something like this: "Overall, ice activation onset conditions between the six samples were similar at all temperatures tested. However, at -40∘C Samples 3 and 4 showed first ice nucleation activity at saturation ice ratios X to X % earlier than the other samples and may be characterized as more efficient deposition INPs at that temperature as compared to the remaining samples. However, it should be emphasized that the presented results are not of a quantitative measure."

Yes, Figure 10 shows the onset conditions of the first observed ice nucleation event. Because this is the first observed ice nucleation event, we can say that our percentage of ice is approximately $10^{-4}$ (1 out of $10^4$; total number of particles on the sample disc) and thus our technique is semi-quantitative. We have changed this to read:

"Overall, ice activation onset conditions between the six samples were similar at all temperatures tested (Figure 10). However, at –40 °C, Samples 3 and 4 showed first ice nucleation activity at a saturation ice ratio of 1.12, which were lower than the other samples and may be characterized as more efficient deposition INPs at that temperature as compared to the remaining samples."

6) My impression is that sections 3.2 and 3.3 are somewhat overanalyzed in terms of trying to explain the sources etc. After all, the result section of this manuscript depicts only six consecutive 30 minute samples of one day, most of which feature a considerable variance in sampling altitude, which should make it difficult to make general statements about the different samples. Also only 3 – 5 INPs were analyzed with Raman spectroscopy from each sampling spot, again making it difficult to allow general statements about each sample. I will give some examples: a) Page 11, Line 10ff: I think this statement is somewhat misleading. Sample 6 had only a short period of approx. 5 minutes where it was hovering close to the ground. The corresponding reasoning is rather speculative. b) Page 11, Line 17f: 1) and 2) are identical, but only phrased differently? c) Page 11, Line 18f: Although it is a logical statement, is this really supported by the data at this point? d) Page 12, Line 5f: It should be avoided to speak too generally here, since only 3-5 particles per sample were analyzed. How this few particles be representative of a whole sample? e) Page 12, Line 15f & Page 12, Line 17ff: This is again rather speculative.

Although Sample 6 was only near the ground for 5 minutes, those 5 minutes had OPC particle concentrations up to 248 cm$^{-3}$. Influence from short exposure to samples for drop freezing analysis can yield significant changes in the freezing temperatures and thus the INP concentrations, which were on a maximum of 0.04 cm$^{-3}$ INPs for Sample 6. The ice nucleation community has this issue with even exposure of blank samples to laboratory air for a matter of seconds to minutes—even small levels of exposure to relatively "clean air" in the laboratory can introduce enough artefacts to increase the background level of even the purest water.

We revised the sentence summarizing the results in section 3.2 to, "Combined, the immersion INP, OPC, and BBFCS results indicate that: 1) total particle number concentrations and INP concentrations were highest when HOVERCAT sampled near the ground and 2) INPs of likely biological origin remained close to the surface, which is predominantly agricultural soils (Hill et al., 2016)." For point 2), this is indeed a logical statement given the freezing temperatures of

the samples—only biological particles have been shown to nucleate ice at temperatures above −15 °C in the atmosphere (see Murray et al. (2012) and Kanji et al. (2017) and references therein).

Murray, B. J., O'Sullivan, D., Atkinson, J. D., and Webb, M. E.: Ice nucleation by particles immersed in supercooled cloud droplets, Chem Soc Rev, 41, 6519-6554, 2012.

Kanji, Z. A., Ladino, L. A., Wex, H., Boose, Y., Burkert-Kohn, M., Cziczo, D. J., and Krämer, M.: Overview of Ice Nucleating Particles, Meteorological Monographs, 58, 1.1-1.33, 2017.

With regard to the Raman section, we dissolved the samples into HPLC grade water, shook and inverted them to ensure complete mixing, and then the sample was nebulized. From this method, we believe that most of the particles on the sample disc will be of similar composition. While we only analyzed 3 – 5 particles, they were always quite similar. We have added a sentence clarifying why we assume that most particles will be of similar composition: "Of the particles that nucleated ice, 3 – 5 particles were analysed for composition using Raman spectrometry for each sample. We assume that a majority of the particles are of similar concentration because the whole sample was dissolved in water, allowed to mix to a homogeneous solution, and nebulized onto the sample disc. Indeed, the particle composition was similar for each particle in any sample, while there was variation from sample to sample."

**Specific/Minor Comments:**

1. The terming of HOVERCATS ability to measure "time-resolved" INP concentrations (Abstract, Page 13 Line 11) seems a little misleading, since it averages over broader time periods instead of measuring in real-time.

The purpose for defining our sampler as time-resolved was to distinguish it from previous balloon-borne measurements where a filter was collected for the entire flight. Time-resolved does necessarily indicate real-time, however, we removed "time-resolved" from most spots in the manuscript for to prevent such confusion. Since this is a part of the TRAPS acronym, we specifically define "time-resolved" when first discussing the sampler in section 2.2

2. Use meters above ground level throughout the manuscript (other than when discussion problems related to the flow of the pump). The Colorado plains has a rather high ground elevation, which may lead the reader to think the balloon was flying higher than it actually was.

Understanding the pressures in which the pump was functional at is an important aspect to the system. We wanted to provide the altitude above ground *and* sea level, to demonstrate that the pump had to work harder at the lower pressures aloft in Colorado as, say, compared to flights in Seattle in which 1000 m a.g.l. would have much higher ambient pressures than 1000 a.g.l. over Colorado. The balloon could certainly fly much higher a.g.l. in lower elevation locations than the 500 m a.g.l. in Colorado. However, for clarity we now ensure that anywhere we state a.m.s.l. that we provide a.g.l. when referring to the flights in Colorado and to be consistent with the a.g.l. in Figure 2, we changed Figure 7 to a.g.l. as well.

3. Page 2, Line 10: When listing the ice nucleation modes, a reference to Vali et al. (2015) should be considered. Also, the concept of pore condensation freezing as introduced by Marcolli et al. (2014) could be mentioned, when describing deposition freezing.

Vali et al (2015) was added, in addition to a mention of pore condensation freezing and the Marcolli et al. (2014) reference.

4. Describe in more detail how the system adjusts the altitude, how it lands and how it is retrieved. Is the balloon still intact after landing, can it be reused?

We added a description of how the balloon system adjusts to the altitude, how it lands, and how it is retrieved in section 2.2. We did not go into detail on how it adjusts to the altitude because we removed discussion on the hovering capability.

5. Why haven't you used a more powerful pump that can go above 1.2 lpm? I understand that payload restrictions are crucial for this approach, but I think the system would be much more flexible when the sample flow could go higher.

It is certainly true that higher flow rates would yield larger volumes of air and thus sufficient sample loading for ice nucleation measurements. However, as our manuscript and the reviewer note, putting a larger pump on the system would indeed exceed the FAA regulations on payload restriction. We are limited to 2.7 kg or less, and our aerosol module weighs 2.3 kg. We have researched a variety of stronger pumps, none of which weight less than 400 g.

6. Page 10, Line 28ff: I think it would be better to give median (or average) particle concentrations instead of the maximum, since near ground values showed episodic peaks. Also, a plot showing the particle concentration vs. altitude could be worthwhile.

We added the averages for when the system was close to the ground versus when it was aloft in section 3.1. We wanted to provide the maximum here to show how influential ground sources are to the particle counts. Figure 7 already demonstrates that when the system was flying near the ground, the OPC counts and mass concentrations clearly increased, thus, a correlation plot would be redundant. Additionally, as indicated by the spikes, the sources of high particle counts were localized at the ground, thus a correlation of such high time resolution data would not be observed. We added this to the discussion in the second paragraph of section 3.1.

7. I wonder if you considered to calculate the correlation between OPC concentration larger 0.5 µm (as in DeMott et al., 2010, 2015) and the INP concentration at T = –X °C averaged over each sampling period (although N = 6 isn't very good statistics), and/or add a scatter plot.

We did look at the OPC concentrations from the third bin to the last (i.e., 540 nm – 17 µm). However, (1) the lower limit can only be 540 nm and not 500 nm and (2) there was no strong correlation with the OPC > 540 nm concentrations and INP concentrations at −10 °C, −15 °C, −20 °C, −25 °C, nor the maximum INP concentration per sample. There is a correlation between the OPC > 540 nm and temperature in which a fraction frozen of 1 occurs, but as the reviewer notes, this is only for N = 6. Thus, we made no such changes in the manuscript. Here is a plot of total OPC concentrations versus concentrations of > 540 nm demonstrating a minor difference in the two:

[Figure]

8. Page 11, Line 12: Give the range of minutes (from X to X minutes) that HOVERCAT hovered near the ground. Following the reasoning on Page 11, Line 14ff it may be worthwhile to correlate the number of minutes close to the ground with the INP concentration of the corresponding sample.

Good idea. We have now added the % of time at the ground for the duration of the sample (i.e., < 50 m a.g.l.) to the discussion in section 3.2. Sample 3 did have the highest percentage of time spent at the ground compared to the other samples (69%), followed by sample 1 (40%), sample 6 (19%), sample 2 (9%), sample 5 (2%), and sample 4 (0%).

9. In section 3.4 the authors should add a goal to improve on the ability to stay at a constant altitude with HOVERCAT.

Done.

10. Add a Figure that shows a detailed schematic of the TRAPS. I see that a similar Figure can be found in Ogren et al. (2017). Yet, I feel it is essential to this manuscript as well and should be added therefore (also the design probably is different to what is shown in Ogren et al. (2017)). A few sentences describing the filter collection with TRAPS in more detail could also added to the text.

Thank you for the suggestion. We have added a schematic of the TRAPS in Figure 1 and have revised the discussion in section 2.1 is sufficient. We also added the pore size of the filter.

11. Fig. 2: Indicate the exact sampling times (e.g. using a black sphere to mark the beginning of a new sample on the flight track, and a black line one the time color scale)

Sampling times are already shown in Figure 7.

12. Fig. 3: Y-axis and color scale give the same information, one of those seems redundant. A stacked (color-coded by cooling rate) histogram might give more information (or as additional Fig. b). Something like my Figure 1 attached to this review (not your data):

[Figure]

Great idea, we agree the histogram looks much better and have changed to that in Figure 3.

13. Fig. 7: Give the altitude in m AGL (starting from 0). Use more distinct colors for altitude and OPC (instead of light and dark grey). Coloring for Sample 2 is not the same orange as in the others figures. Maybe use logarithmic scale for the OPC concentration?

Done.

14. Fig. 10: Add information to the labeling describing what exactly is depicted on this figure. Is it the onset conditions of the first observed ice nucleation event (or of X % activated fraction)?

Thank you for the suggested clarification, we have changed the caption to read: "Depositional ice nucleation experiments on Samples 1 – 6 plotted by $S_{ice}$ versus temperature. The values plotted here are of the onset conditions of depositional ice nucleation. For our experiments, this refers to the first particle to nucleate ice out of the $10^4$ particles deposited on the disc in total, thus a percent activated fraction of $10^{-4}$. Although temperatures measured were not exactly −25 °C, −40 °C, and −55 °C, these values are used for brevity for all samples within each grouping shown above."

15. Fig. 11: What is meant by "most representative particle type per sample"? In section 2.3.2 it says that only 3 – 5 particles were analyzed with Raman spectrometry per sample. How do you know which particle type is representative for the whole sample when only 3 – 5 particles each were analyzed in total?

We have changed the wording here because we meant that the spectrum for each sample is representative of the entire sample. The particles on the disc had little to no variation in spectral features, which is expected from the sample preparation. The solutions that were nebulized were homogeneous. We added the following to clarify this: "Of the

particles that nucleated ice, 3 – 5 particles were analysed for composition using Raman spectrometry for each sample. We assume that a majority of the particles are of similar concentration because the whole sample was dissolved in water, allowed to mix to a homogeneous solution, and nebulized onto the sample disc. Indeed, the particle composition was similar for each particle in any sample, while there was variation from sample to sample."

The caption for Figure 11 has been modified to read: "Raman spectra for a representative particle per sample. Characteristic vibrational frequencies for functional groups of organics (C-H; $2800 – 3000$ $cm^{-1}$), carbonates ($CO_3$; $1070 – 1090$ $cm^{-1}$), sulphates ($SO_4$; $972 – 1008$ $cm^{-1}$), and nitrates ($NO_3$; $1032 – 1069$ $cm^{-1}$) are noted for reference. Included are images of the particles that initiated depositional freezing for the Raman spectra shown. The length of the black line in each image represents a scale of $20$ μm."

**Technical Comments:**

Page 2, Line 17: Replace "adroit" with "efficient"

Done.

Page 5, Line 10: Remove "-" in "1.2-L min-1"

Done.

Page 5, Line 13: Add space between "12 V DC"

Done.

Page 10, Line 23ff: The term "profile" is a little confusing, since it does not correspond to the same samplings (e.g. profile 3 is sample 4). Maybe add a short sentence that the two terms are not the same.

We now define profile as "ascent followed by descent to ground" as provided in the first paragraph of section 3.1.

Page 12, Line 16: Typo in reference "Möhler et al., 2008"

Fixed.

Fig. 1 Caption: b) Picture of the aerosol module.

Fixed.

Fig. 8: Labeling of "UPW in breaker" is covering part of the data. The last sentence of the caption does not make sense for this plot.

Fixed labeling and removed last sentence of caption.

[revised manuscript text omitted]

---

## Author Comment (AC2) · 23 May 2018

We would like to thank the reviewers for their insightful and helpful comments, and encouragement for the need of vertical profiling of INPs. As a result of revision based on their feedback, the manuscript is much stronger. Below are the responses to reviewer comments in blue. We note that both reviewers were generally concerned with the hovering capability of the system in addition to presenting one successful flight out of three. As a result, we have significantly revised the tone and organization of the manuscript, in addition to providing the community with what did and did not work, and recommendations for those interested in developing similar systems in the future. More details and specific examples of where text was modified is provided in the responses below. We have attached a track changes version for reference of the specific revisions made.

**Reviewer 2**

**Page 1**
Line 11. Precipitation processes "can" be modulated by aerosol cloud, not "are" since other dynamics also influence radiative and precipitation processes.

Fixed.

Line 15-20: Upon reading the manuscript and seeing how only 1 successful test has been presented herein, I would soften the ideas about "hovering" at as given altitude. It was clear that the authors struggled to hover at a given altitude because of vertical winds and the balloon could not respond fast enough to maintain altitude. I think one test flight is too soon to claim this.

The reviewer has a valid concern, similar to the first reviewer. As described at the beginning of this response, we have significantly revised the manuscript to soften the "hovering" aspect of the work, given the significant and unexpected challenges associated with such an idea situation, and instead highlight the utility of the aerosol module and evaluation of vertical distributions of INPs.

For realtime aerosol measurements, the vertical profile would still be achieveavle, but for INP analysis as the authors rightfully acknowledge more time is needed at each sampling altitude to get enough PM on their filter, and this is harder to achievbe without fluctuations (sudden rises, or falls). Perhaps it would help if the authors stated a range xx +/- yy m hovering capability. So that instead of being sample to sample at say 500 M agl.. HOVERCAT can sample at 500 +/-100 m as an example.

We have softened the hovering aspect of the system throughout the manuscript to rectify the issue with the hovering flight plan. We also added, "However, this plan was ultimately not executed due to flight complications discussed herein." to section 2.4. The only places the word "hover" remains in the manuscript is in the HOVERCAT acronym. We also revised the definition of the BBFCS system such that the user can control the ascent and descent of the balloon instead of "controlling the altitude".

To provide some specific examples of where we modified the text:
-   We revised the abstract to highlight the aerosol module and state that we provide recommendations for use of future launched platforms.
-   We instead emphasize that the BBFCS has a benefit of being more cost effective and flexible (in terms of FAA regulations, launch locations, and personnel required) than traditional tethered platforms such that it does not require a winch and that we can slow the ascent rate in addition to providing a descent option which, combined, is not a feather attainable with traditional launched balloons. This information is provided in section 2.1.
-   We completely revised and reorganized the methods section to clearly define HOVERCAT and describe that first, while describing but not showcasing the balloon platform.
-   We revised section 3.4 to emphasize our scientific priorities (discussed in detail in our response to comment 2) and recommendations for future measurements.

Where we do discuss particular altitudes that were somewhat maintained (i.e., in section 3.1) we now provide a ± range.

**Page 2**

line 10. An appropriate reference to add or even to replace Cziczo et al. (2017)(who talk about IN methods as a focus not the mechanisms) with would Vali et al. (2015) who have a nice discussion that included the IN community on the terminology and IN mechanisms.

We have added Vali et al. (2015).

Line 12: immersion freezing is the most relevant heterogeneous ice nucleation mechanism for MPCs but not for cloud ice formation. We don't know that, there is a fair bit of literature to suggest that majority of cloud ice may come from secondary ice multiplication. So I suggest re-phrase to make this more clear.

Good point. We have revised the sentence to, "Immersion freezing is the most relevant to primary ice formation in mixed-phase clouds…"

**Page 3**

Line 1: here the authors say 2 km AGL is a limitation but then the platform they present is only shown to go to 1.1 km AGL, this is contradictory. I suggest correct this or modify the way this is presented.

We assume the reviewer is talking about the top of page 4, but these are discussing limitations of tethered balloons. We do see the point that we only go up to 1.1 km a.g.l. in Colorado, but this limited by the low pressures associated with this altitude. If, for example, we launched in Seattle, we could go much higher than 1.1 m a.g.l. However, to eliminate any confusion and focus more on the other caveats with tethered balloons, we now state, "In general, tethered balloons can handle much larger payloads than launched systems, but are limited to lower altitudes (i.e., up to approximately 2 km a.g.l. anywhere), have wind condition limitations, and involve more complicated logistics (e.g., use of a winch and personnel required to operate a winch) thus may not be ideal for sampling at various levels where clouds exist."

Line 4 -5: there may have not been INP measurements on a balloon system, but a similar idea of collecting aerosol on filters/wafers on a UAV was already done couple years ago and published (Schrod et al., 2017). This study should be discussed and acknowledged here.

Reviewer 1 also brought to our attention Schrod et al. (2017) and Ardon-Dryer et al. (2011), which were actual INP measurements from balloon. We have added both Schrod et al. (2017) and Ardon-Dryer et al. (2011) to the introduction, their advantages, and how our results differ from theirs. Schrod et al. (2017) measured INPs by only in the deposition mode. Ardon-Dryer et al. (2011) present results more relevant to ours, but only measured INPs below −18 °C in immersion mode on 3 samples and only up to 196 m a.g.l. The fact that only two studies before ours exist, and the limitations of those studies and ours, demonstrates the challenges associated with obtaining vertically profiled INPs, which is important to highlight and why we directly state this after describing their work in the introduction. We also removed the statement, "Additionally, to our knowledge, there have been no measurements of INPs via any balloon platform."

Ardon-Dryer, K., Levin, Z., and Lawson, R. P.: Characteristics of immersion freezing nuclei at the South Pole station in Antarctica, Atmos Chem Phys, 11, 4015-4024, 2011.

**Page 5**

Line 8: Maybe state what density/shape factor relationships were used to convert OPC data into PM values

We added the density and refractive index values used (default of 1.65 g mL$^{-1}$ and 1.5, respectively).

Line 10: typo should be 1.2 L not 1.2-L

Fixed.

**Page 6**

This is a nice description of the flight and the testing done. Indeed, it is exciting to read that such developments are taking place. My only concern about this is if two out of the three test flights were not successful and if the altitude of 2.5 km AMSL was determined to be the max altitude range of operation, why weren't more flights conducted for this study, once the operational parameters were clear? This should have been straightforward to do, no? Furthermore, if stronger pumps are being tested for higher altitude (lower pressure) sampling why not wait for those tests so that a test flight with higher altitude can also be conducted.

To address the issue with having only one successful flight, we note that the system *does* work, but with its caveats. Even Ardon-Dryer et al. (2011) that reviewer 1 mentions (i.e., the only other INP measurements by balloon in existence) present only three filters from one day in their paper, indicating a substantial number of flights and samples is not required for publication of what (very) limited INP profiles exist. We emphasize that this is Phase I and think it is important to discuss what *didn't* work as that is important to other researchers who may be interested in developing a parallel system in the future, such that they will not have to "reinvent the wheel" so to speak. This is the reasoning behind keeping the original flight plan, such that our idea of stepping is not as easy as one might think. However, we now modified section 3.4 to discuss our recommendations for those interested in controlled launched balloon measurements, including a list of what to do and not to do. This addition provides rationale for keeping in the flight plans, why we had one out of three successful flights, and provides suggestions for improvement. There is a certain stigma with publishing only partially successful results or deployments, but we aim to break that mold and show what didn't work to help those in the future. In addition, we originally intended to showcase the vertical profiling of both immersion and deposition mode INPs with the aerosol module, which is indeed novel, but have now emphasized that throughout the manuscript.

It is certainly true that higher flow rates would yield larger volumes of air and thus sufficient sample loading for ice nucleation measurements. However, as our manuscript, putting a larger pump on the system would indeed exceed the FAA regulations on payload restriction. We are limited to 2.7 kg or less, and our aerosol module weighs 2.3 kg. We have researched a variety of stronger pumps, none of which weight less than 400 g. We have modified the text in section 3.4 to explain that a larger pump could be used, but likely only on tethered balloon platforms or UAS that can hold a large payload.

Line 22-28: The data given in this paragraph is all for ground level or at a given height? I guess if the goal is vertical profiling, shouldn't the meteorological data for the different height be useful or may be needed to understand the aerosol properties (e.g. phase state) during sampling. Does it help to have all the ground based data when all the sampling is occurring above ground?

As noted, the data were obtained from the CDPHE at their Boulder Reservoir site, although we realize we did not directly state these data were from a ground site, which we now provide. To our knowledge, there are no meteorological data available above the ground near the locations we flew. The goal of describing the general conditions was to paint a picture of the overall weather that day.

**Page 7**

Line 1: make clear 47 mm is diameter.

Fixed.

Line 25: here the authors say control experiments were done with ultrapure water at various cooling rates. How was the water treated here. In order to have a true control, the water should also have been subjected to the same process as the samples i.e. " (i.e., samples) were successfully collected before the battery died. Each spot was placed is a 29-mL sterile Whirlpak® bag with 2 mL of UPW to resuspend particles deposited on the filter" The water should be treated as above to ensure there is no INP contamination coming from the whirlpak bag. Have such experiments been done? If not, I suggest these should be included in order to really be confident that the freezing is from the aerosol re-suspended into the sample.

The water was from a commercial ultrapure water system as described in the methods, the same UPW that was used for the samples. We did not conduct additional treatment to the UPW. Additionally, we did include spectra in Figure

8 of UPW alone in a beaker, UPW mixed in a WhirlPak® bag for 2 hours, and the filter used (EmFab®) mixed in UPW in a WhirlPak® bag for 2 hours. Thus, we did present results of blanks treated in the same manner as the samples. However, we made this clearer in the caption of Figure 8.

**Page 8**

Line 4: I understand how the temperature calibration experiment was done, but I tried hard but do not follow how the data are plotted in Figure 3. Which data are for the probe at the center of the copper plate and which for the drop on top of the stage? The vertical lines just mean that the temperature difference doesn't change with cooling rate, but then why are there so many vertical lines, and what determined the position of the data on the x-axis, this figure is difficult to follow. I also don't understand why there is a cooling rate on the y-axis and then the data are also colored by cooling rate?

We have completely revised Figure 3 to show a histogram and eliminate redundant information.

Line 27, why does it take more time to prepare drops with a pipette. Are you using a single tipped pipette, but you could use an 8-channel or 12 channel pipette for this too. It would speed things up.

It takes more time because it is a single pipette instead of a repeater (which we do not have access to). With the normal pipette, we have to draw and expel sample for each drop versus the syringe, which is filled with the whole 0.25 mL then dropped on the plate. A multichannel pipette is a good idea in theory, but would not allow for the circular pattern of drops on the copper disc and would create a much smaller number of drops (we have already tried this and it did not work).

**Page 9**

Line 4. It was not clear to me why all 100 drops could not be recorded? If you place 100 drops on a cold stage what is the hindrance in recording data until all 100 drops are frozen?

The limitation of the DFCP is the hindrance: At times, not all drops froze for the control experiments with UPW due to the system detection limit (i.e., it can only go down to approximately −32 to −33 °C). Not all water freezes by these temperatures, so we had times where a few drops remained unfrozen. We more clearly define this in section 2.3.1.

**Page 10**

Line 10: based on your filter sampling, what was the efficiency of smapling particles larger than 10um? From the deposition mode experiments the authors say they see particles between 1 – 50 um. This could be a strong indication of coagulation of the aerosol based on the suspension and nebulization methods used. Does it then make sense to assume that one particle lead to each ice crystal observed? And also try and analyses that particle? It is likely a particle that is composed of multiple particles. With more samples perhaps the authors could do a particle count by feeding the nebulized spray though an SMPS or even into a CPC to get a particle count. And then compare this to a particle count taken from image analysis of the deposition state to see if numbers are similar. This will provide some correction factor (if needed) for how much coagulation maybe taking place when the particles are being deposited onto the stage for deposition nucleation.

According to Ogren et al. (2017), there is an 80% collection efficiency of 10-µm particles, but according to their Figure 2, the instrument can collect at larger sizes, just at a lower efficiency. However, it is important to note that the sampling efficiencies were calculated, and not actually measured. We have added this information to section 2.1, stating, "The TRAPS has the highest collection efficiency for particles in the 1 nm – 10 µm aerodynamic diameter range—with particle losses of less than 10% for 5 nm – 7 µm particles and less than 1% for 30 nm – 2.5 µm particles at 1.0 L min-1—but can collect particles with larger diameters (Ogren et al., 2017)."

[Figure]

**Figure 2.** Calculated sampling efficiency of particles reaching the CLAP filter.

A more detailed experimental setup would be ideal, however, due to the limited amount of sample, we were not able to run extra experiments with size distribution measurments. We did add some text describing the caveats "Although the Raman spectral and ice nucleation analyses are helpful to observe the overall particle composition as temperature and relative humidity are changed, the experiment does not determine the size or mixing state of the particles as they were in the atmosphere. Further, the spectral resolution of 1 micrometre in our system does not allow smaller scales to be distinguished within the individual particles probed."

Line 25-30: regarding sudden drops and hovering – see comment above that I made in abstract section. I think it is too soon to claim the ability to hover at a fixed altitude. Perhaps consider putting a range to that, given that the balloon cannot respond fast enough to the updrafts experienced. I still think this method is super valuable even for example you say, each INP filter sample is conducted within a range of 100 m or so. This is a step in the right direction for vertical profiling

We completely agree with the reviewer that is it too soon to claim the ability to hover. We have significantly revised the manuscript to avoid this issue as discussed in our response to the reviewer's last comment on page 1.

**Page 11**

Section 3.2. Results are indeed interesting, but it would have been nice to see that each sample presented did not have overlapping altitudes. Is there no way to automatically shut off filter sampling when a certain threshold of a sudden rise or fall is crossed and then only resume sampling when the balloon is within this threshold again? Would this be too demanding, or is there not enough time in the flight for this. This way you ensure that your INP filter samples are restricted to sampling at a certain altitude and you can resolve better the vertical profiling of the INPS. For example in Figure 9, sample 2, if the sampling was shut off for the periods when the instrument was below 600 m AGL, that would allow sample 2 to be representative of only INPs above 600m AGL.

This is certainly a feature of interest, but extensive work would need to be executed to incorporate bidirectional communications yet maintain payload weight under the FAA regulations. It would be ideal to enable ground-based bidirectional commands to change the sample spot when desired. However, the manuscript presents a pilot study focusing on specific priorities: (1) Can we recover the system? And (2) Can we control the ascent and descent? Both priorities fall under the umbrella of making a system that is relatively affordable, user friendly, and ensuring avoidance of complicated FAA paperwork by obtaining a COA. Adding complexity to the system for controlling altitude and sampling frequency adds cost, weight, and more variables that could fail. For clarity, we now emphasize that this was a pilot study throughout the manuscript (in addition what we already call "test" flights) and discuss our priorities in section 3.4.

More specifically, a bidirectional communication feature entails a significantly more advanced ground-based system and would add too much weight to HOVERCAT through additional processing hardware, an additional receiver, and batteries needed to control in such a manner. First, the BBFCS had bidirectional communication, but additional channels would need to be added to control the TRAPS as well, adding to the cost. And second, adding the necessary

components to the TRAPS would add to both cost and weight, and given our 400 g buffer (i.e., calculated from the 2700 g limit and our HOVERCAT 2300 g weight), would likely push the system into a weight category that requires either restricted airspace or significant paperwork to obtain and FAA COA to fly outside of restricted airspace areas. Both of these issues make flying HOVERCAT more complicated and expensive, which fall outside of our vision to make an affordable and easy-to-use system.

Section 3.3: should be deposition nucleation – not freezing, freezing implies liquid to solid transition, I think the authors' intent is to imply vapour to ice mechanism here, therefore I would take freezing out of the phrase.

Thank you for pointing this out. We have changed 'deposition freezing' to 'deposition nucleation' throughout the manuscript.

**Page 12**

Line 1: "…above homogenous freezing" are you referring to temperature or Si? Could clarify here already. Based on Fig. 10 it looks like you refer to both. However, some of your data points are quite close to water saturation (-25 for example), so it maybe a tough sell to claim those as be depositional ice nucleation.

Thanks for that catch. We mean the $S_{ice}$ values are below that required for homogeneous freezing. For the lower temperatures, we are confident that we are observing depositional ice nucleation, but for the highest temperature it could be subsaturated immersion nucleation. This text was modified from the original version and we removed that statement.

Section 3.3: I like the discussion in this section. But here one must acknowledge that there could be artifacts of having your INPs as agglomerates of particles. So maybe the aerosol was externally mixed, but with the experimental method used, they could become or appear as internally mixed since you suspend and nebulize and then allow for evaporation to retain the residual particles on the cold stage. This should allow for some coagulation, and perhaps the Raman is investigating a single particle that is a result of multiple smaller coagulated particles.

We have added the following to address this: "Although the Raman spectral and ice nucleation analyses are helpful to observe the overall particle composition as temperature and relative humidity are changed, the experiment does not determine the size or mixing state of the particles as they were in the atmosphere. Further, the spectral resolution of 1 micrometre in our system does not allow smaller scales to be distinguished within the individual particles probed."

Section 3.4: Again while reading this section, I can't help but feel that the paper is a little premature. Some of the plans and developments described herein could be already part of this manuscript, like the stronger pumps, a few more test flights, certainly more than 1, and the practice to control the balloon to stay at desired altitudes. i.e., the above should be part of Phase I. I understand regulatory approval work like compliance with FAA can be phase II as well as launching on other airborne platforms (UAS, reverse parachute). i.e. demonstrating their instruments are versatile enough for other platforms, can all be phase II.

We have completed overhauled this section to discuss possible improvements and recommendations for future flights as discussed in the responses above. We realize that there are improvements that could be made, but we aim to clearly highlight our Phase I scientific objectives and questions, by starting simple and developing a system that can be deployed under limited resources. We hope by shifting the focus of section 3.4 to our recommendations that others can help contribute to improvement of the technology and contribute to INP measurements via balloon, which are extremely limited.

**Page 13**

Line 5. Certainly if the plans to operate at Jungfraujoch in Spring 2018 are on track (that is now), then stronger pumps have been implemented already. Since ground level at Jungfraujoch is already ~ 650 mb pressure.

We removed this part as deployment on Jungfraujoch was not executed.

Line 14: maybe put in also the altitude AGL to give context to your starting point.

Done.

Line 19-20: Soften the vertical profiling statement here because the authors showed overlapping vertical profiles, so it is not yet achievable in the strictest of sense.

Done.

Given that this is an instrument development paper, I would like to see a section on Benefits and limitations of the instrument. Where the authors state this clearly. I believe some of this is interspersed through the manuscript, but I think this should be brought together in one section to make it clear what are the benefits of HOVERCAT (and its current limitations).

Done. We have made this information more clear in section 3.4.

Figure 4. For clarity that the control experiments are done with pure water and also it would be good to indicate the volume of the drops in the caption.

Done.

Figure 5. I was a little confused – the "by hand" drops, are those not the same as the syringe drops? Or how were the 2.5 ul drops by hand produced? There must have been some sort of tool for these.

The 'by hand' is technically 'by syringe', which we now changed in the legend and caption for clarity.

Figure 7: Could you make it clear in the caption that the grey line plot is the one corresponding to the altitude and the scatter plot corresponds to the concentration from the OPC? Is it mentioned what densities or shape factors have been assumed to come up with the PM values from the OPC data?

We have revised this figure to better distinguish altitude and OPC number concentrations. We also now provide the density and refractive index used to estimate the PM concentrations.

Were the particles losses of 10% accounted for in the results? i.e. when calculating the concentrations in Fig. 8. Also, how were the control experiments for the ultra pure water in beaker and bag converted to INP concentrations? These fraction curves are not addressed sufficiently in the text I think.

We did not account for particle losses of 10%, because according to Ogren et al. (2017), losses vary depending on particle shape, which is difficult given we do not know the size distribution of the particles collected on our filter.

We now include discussion at the end of section 2.3.1 about how INP concentrations from the blanks were obtained. We assumed an average volume of air based on the samples collected and applied that to the Vali (1961) equation.

[revised manuscript text omitted]